# KMT5C leverages disorder to optimize cooperation with HP1 for heterochromatin retention

Justin W Knechtel [ID] [1], Hilmar Strickfaden[1], Kristal Missiaen[1], Joanne D Hadfield[1], Michael J Hendzel [ID] [1,2] & D Alan Underhill [ID] [1,3] [✉]

## Abstract

A defining feature of constitutive heterochromatin compartments is the heterochromatin protein-1 (HP1) family, whose members display fast internal mobility and rapid exchange with the surrounding nucleoplasm. Here, we describe a paradoxical state for the lysine methyltransferase KMT5C characterized by rapid internal diffusion but minimal nucleoplasmic exchange. This retentive behavior is conferred by sparse sequence features that constitute two modules tethered by an intrinsically disordered linker. While both modules harbor variant HP1 interaction motifs, the first comprises adjacent sequences that increase affinity using avidity. The second motif increases HP1 effective concentration to further enhance affinity in a context-dependent manner, which is evident using distinct heterochromatin recruitment strategies and heterologous linkers with defined conformational ensembles. Despite the linker sequence being highly divergent, it is under evolutionary constraint for functional length, suggesting conformational buffering can support cooperativity between modules across distant orthologs. Overall, we show that KMT5C has evolved a robust tethering strategy that uses minimal sequence determinants to harness highly dynamic HP1 proteins for retention within heterochromatin compartments.

**Keywords** Lysine Methyltransferase 5C (KMT5C); Heterochromatin Protein-1 (HP1); Protein Compartmentalization; Protein Dynamics; Intrinsic Disorder
**Subject Category** Chromatin, Transcription & Genomics

## Introduction

Chromatin is physically and functionally compartmentalized across a range of length scales (Bhat et al, 2021; Hildebrand and Dekker, 2020; Rowley and Corces, 2018). For instance, the genome is broadly separated into transcriptionally permissive and repressive compartments through the formation of euchromatin and heterochromatin. The latter is further categorized as facultative or constitutive, which occupy distinct spatial domains within the interphase nucleus. This is particularly evident in mouse cells where pericentromeres from multiple chromosomes congregate into larger domains called chromocenters (Haaf and Schmid, 1991; Guenatri et al, 2004). These structures are primarily comprised of non-coding repetitive DNA and characterized by trimethylation of histone H3 lysine-9 (H3K9me3) and histone H4 lysine-20 (H4K20me3), as well as enrichment of heterochromatin protein-1 (HP1) family members (HP1α/CBX5, HP1β/CBX1, and HP1γ/CBX3) (Nishibuchi and Dejardin, 2017; Ostromyshenskii et al, 2018; Saksouk et al, 2015). This is thought to occur in a stepwise manner involving catalysis of H3K9me3 by the lysine methyltransferases SUV39H1 and 2 (Peters et al, 2001) (aka KMT1A and B), recognition of H3K9me3 by HP1 proteins, and subsequent recruitment of the lysine methyltransferase KMT5C (ak*a* SUV420H2) to catalyze H4K20me3 (Schotta et al, 2004).

KMT5C localization to chromocenters is dependent on direct interaction with HP1 proteins (Hahn et al, 2013; Schotta et al, 2004; Souza et al, 2009). Nevertheless, they exhibit dramatically different bulk dynamics within and from heterochromatin compartments. While HP1 members exchange rapidly between the nucleoplasm and chromocenters when assessed by fluorescence recovery after photobleaching (FRAP) (Cheutin et al, 2003; Muller-Ott et al, 2014), studies of KMT5C have suggested it formed an immobile scaffold (Hahn et al, 2013; Muller-Ott et al, 2014; Souza et al, 2009). This conclusion was supported by analyses in mouse embryonic stem cells (Hahn et al, 2013) and L929 (Souza et al, 2009), NIH 3T3 (Muller-Ott et al, 2014), and mouse embryonic fibroblasts (Hahn et al, 2013; Muller-Ott et al, 2014), supporting its generality. Moreover, it fits with a role for KMT5C in heterochromatin compaction (Hahn et al, 2013) and that its catalysis of H4K20me3 was under strict spatial (Biron et al, 2004; Martens et al, 2005; Schotta et al, 2004) and quantitative (Leroy et al, 2013; Pesavento et al, 2008; Schotta et al, 2008) control. Nevertheless, an overlooked component of previous analyses of chromocenter protein dynamics relates to the mobility of molecules within these compartments (Muzzopappa et al, 2022). This is particularly relevant given the ability of HP1 to undergo liquid-liquid phase separation (Larson et al, 2017; Strom et al, 2017), which has the capacity to

[1]Department of Oncology, Faculty of Medicine & Dentistry, University of Alberta, Edmonton, AB, Canada. [2]Department of Cell Biology, Faculty of Medicine & Dentistry, University of Alberta, Edmonton, AB, Canada. [3]Department of Medical Genetics, Faculty of Medicine & Dentistry, University of Alberta, Edmonton, AB, Canada.
[✉]E-mail: alan.underhill@ualberta.ca

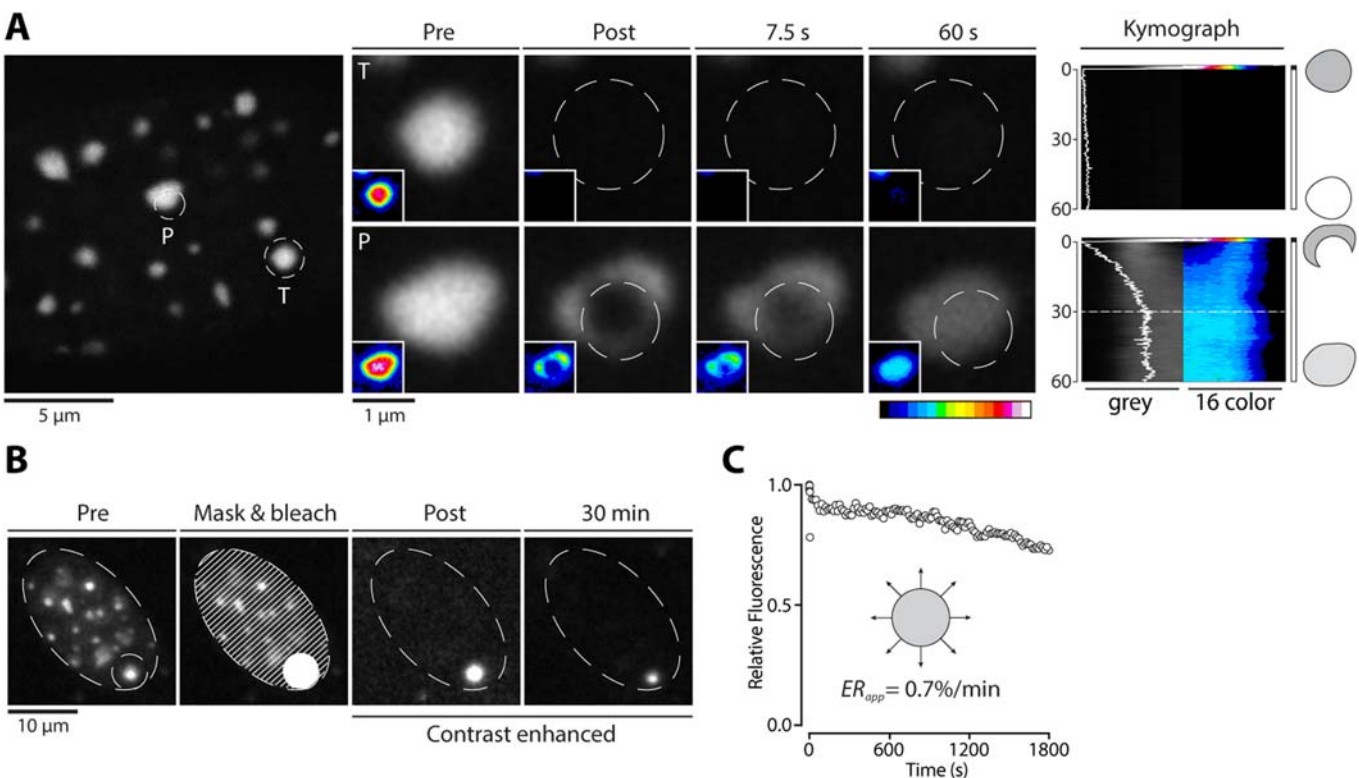

**Figure 1. Characterization of KMT5C dynamics within constitutive heterochromatin compartments.**

(A) Time-lapse series of total (T) and partial (P) transiently transfected KMT5C-mEmerald fluorescence recovery after photobleaching (Movie EV1) in mouse NMuMG immortalized breast epithelial cells ($n = 45$) (see Methods). Representative images are shown for pre-bleach, post-bleach (0.3 s), 7.5 s, and 60 s. Kymographs show fluorescence prior to photobleaching and over the complete 60-s time series (indicated schematically on the right) with the corresponding intensity profile. Insets show fluorescence intensity using 16-color LUT. (B) Inverse FRAP (iFRAP) time series depicts the pre-bleach state, immediately after photobleaching the entire nucleus except for a single chromocenter, and at 30 min. (C) Relative fluoresce as a function of time is plotted on the right, which revealed an apparent efflux rate ($ER_{app}$) of 0.7%/min. Source data are available online for this figure.

influence partitioning and dynamics of heterochromatin-associated proteins (Banani et al, 2016; Ditlev et al, 2018).

In the present study, we assessed inter and intra-chromocenter mobility for KMT5C using FRAP analyses, which, counter to expectation, revealed that KMT5C was not stably bound but sampled the entire chromocenter volume on the timescale of seconds. Phylogenetic and functional analyses established that this retentive activity was consolidated within a discrete protein segment comprised of two short modules connected by a highly variable intrinsically disordered linker. We then leveraged sparse sequence conservation of this region to guide systematic deletion and mutagenesis to map drivers of KMT5C partitioning and dynamics, and their dependence on HP1 interactions (Hahn et al, 2013; Souza et al, 2009). Together with altering the physicochemical properties of the disordered linker, it was possible to develop a mechanistic model for KMT5C retention within chromocenters. The model highlights key roles for the disordered linker in modulating effective concentration and sensing of the chromatin environment, and that precise spatial control of protein localization can be achieved with astonishingly few sequence features. Overall, this provides a paradigm to test the limits of protein compartmentalization in the absence of membranes and the underlying design principles.

# Results

## KMT5C defines a novel dynamic state within constitutive heterochromatin

In previous FRAP studies, CBX5 rapidly exchanged between chromocenters and the nucleoplasm (Cheutin et al, 2003; Muller-Ott et al, 2014; Festenstein et al, 2003; Souza et al, 2009), whereas SUV39H2 and KMT5C appeared to be immobile when entire chromocenters were bleached (Hahn et al, 2013; Muller-Ott et al, 2014; Souza et al, 2009). Although concluding the latter two proteins were stably bound, these studies did not assess mobility within chromocenters. To evaluate the possibility of KMT5C mobility within constitutive heterochromatin, we queried KMT5C dynamics using a combination of total and partial photobleaching of chromocenters. As shown with total chromocenter bleaching (Muller-Ott et al, 2014; Hahn et al, 2013; Souza et al, 2009), KMT5C displayed long recovery times, indicating minimal exchange with the surrounding nucleoplasm (Fig. 1A, Total; Movie EV1). Upon partial bleaching, however, fluorescence recovered (Movie EV1) and progressed from the non-bleached portion of the chromocenter without leaving a residual phantom (Fig. 1A, Partial). This occurred in ~30 s (Fig. 1A, kymograph) and

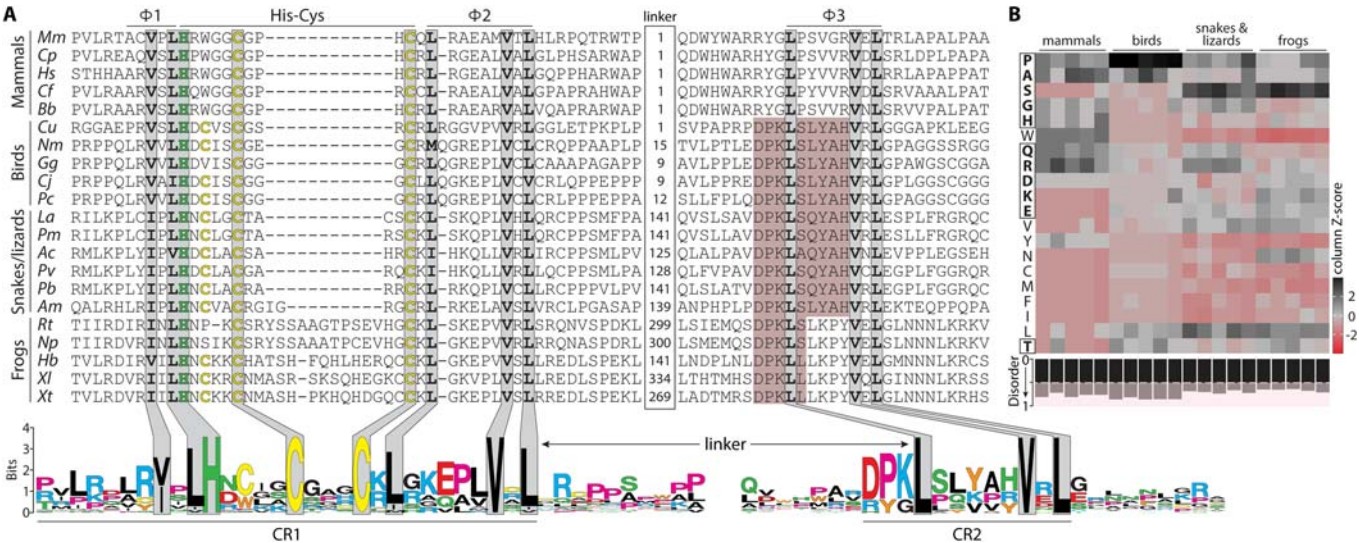

**Figure 2. Sequence features of the KMT5C heterochromatin retention domain.**

(A) Alignment of representative heterochromatin retention domains (HRD) from mammals, birds, reptiles (lizards and snakes), and frogs. Conserved residues that make up each of the four sequence motifs (Φ1, His-Cys, Φ2, and Φ3) are highlighted in gray. A portion of the linker region is shown with the remaining number of residues indicated (linker). The extended region of homology overlapping with Φ3 in birds, reptiles, and frogs is shown in pink highlight. A gapless version of the same sequences is shown below as a WebLogo (Crooks et al, 2004) to illustrate sparse identity and the two conserved regions (CR1 and CR2). (B) Linker amino acid composition is shown as a heatmap for representative mammals, birds, reptiles, and frogs with the scale denoting column z-score. Disorder-promoting residues are boxed, and the overall disorder propensity for each linker is provided in the inverted bar graph below.

suggested KMT5C fully remixed within the heterochromatin compartment with no apparent immobile fraction. Over the same timescale, there was no appreciable fluorescence recovery of fully bleached chromocenters in the same nucleus, together establishing that KMT5C moved readily within chromocenters but did not efficiently exchange.

The retentive nature of KMT5C localization was particularly evident in inverse FRAP experiments where a single chromocenter was spared from photobleaching, and its fluorescence intensity was measured over 30 min (Fig. 1B). In the absence of fluorescent protein influx, this revealed prolonged maintenance of KMT5C-mEmerald signal intensity and an apparent efflux rate of $0.7\%\,min^{-1}$ (Fig. 1C). By extrapolation, complete chromocenter turnover of KMT5C would require ~2.4 h, which corresponds to a 290-fold difference in equilibration rate in comparison to internal recovery. This slow rate of exchange would not be apparent over the shorter 100 s time course used in our conventional FRAP analyses. Overall, the data show that KMT5C equilibrated within individual chromocenters with minimal efflux to the nucleoplasm such that internal mobility and exchange occurred on dramatically different time scales.

## Limited sequence features confer KMT5C chromocenter retention

We sought to understand the driver(s) of KMT5C dynamic behavior through characterization of the underlying sequence determinants. In this regard, sequence requirements for KMT5C heterochromatin localization and dynamics have been mapped to discrete segments of the human (347–435) (Souza et al, 2009) and mouse (350–412) (Hahn et al, 2013) proteins. These segments showed no obvious homology to the paralogous KMT5B protein (*aka* SUV420H1). Using phylogenetic comparisons in mammals, we refined the boundaries of this region to amino acids 354–409 in human KMT5C (356–411 in mouse). This segment recapitulated the behavior of full-length KMT5C regarding localization, internal chromocenter mobility, and lack of nucleoplasmic exchange (Fig. EV1; Movie EV2). To reflect the activities of this short region, we adopted the term <u>h</u>eterochromatin <u>r</u>etention <u>d</u>omain (HRD), although it had previously been called the heterochromatin targeting module (Souza et al, 2009) or clamp domain (Hahn et al, 2013). Sequence identity of this region amongst mammals, however, was limited to 17 of 57 residues (29.8%), and there was considerable variation within individual orders—for example, 21 positions were variable within Primates. Through manual annotation in KMT5C orthologs (Grau-Bové et al, 2022), we identified HRD counterparts in birds, reptiles, and amphibians, as well as more distant eukaryotes (Fig. 2A, top). When these sequences were consolidated (Fig. 2A, bottom; Fig. EV2), the HRD could be defined based on the presence of four short <u>lin</u>ear motifs (SLiMs): a V/I-x-L hydrophobic motif (Φ1), a His-$x_4$-Cys-$x_{3-4}$-Cys motif (His-Cys), a L-$x_{5-6}$-V-x-L hydrophobic motif (Φ2), and a related a L-$x_5$-V-x-L hydrophobic motif (Φ3). The corresponding His-Cys motif in Amphibia was longer and more variable (His-$x_{3-4}$-Cys-$x_{13-14}$-Cys). In all cases, the spacing between the Φ1, His-Cys, and Φ2 motifs was invariant (0 and 1 residue, respectively) across and within taxonomic classes (Fig. 2A, top), indicating their architecture was under evolutionary constraint. The Φ3 motif was separated by a segment that was highly variable in length and amino acid composition but showed enrichment for disorder-promoting amino acids (Fig. 2B) (Uversky, 2013). We referred to the two modules flanking the linker as conserved regions 1 (CR1) and 2

(CR2) (Fig. 2A, bottom), which fit with previous analyses of KMT5C where two adjacent regions cooperated to confer chromocenter localization (Hahn et al, 2013; Souza et al, 2009). Linker length between CR1 and CR2 in placental mammals was constant at 21 residues and increased progressively in birds (21-35 residues), reptiles (145–169), and frogs (>300) (Fig. 2A,B). While this made the identification of Φ3 challenging, it was generally embedded within a short region of class-specific homology (Fig. 2A, top). Moreover, for the bird, reptile, and frog Φ3 derivatives, there was a subregion of amino acid identity (Fig. 2A, top, pink background) that suggested a common evolutionary origin even with substantive differences in the surrounding sequence and linker (Fig. 2B). Collectively, this sequence annotation created a framework for defining the mechanistic underpinnings of HRD activity that focused on the individual SLiMs, their combinatorial use in CR1 and CR2, and the role of the variable linker.

## Retention is achieved through high cooperativity between SLiMs and unmasking of latent binding activity

The sparse sequence identity of the HRD enabled systematic perturbation of conserved motifs. To simplify their description, individual motifs were assigned a number or letter: 1 (Φ1); H (His); C (Cys); 2 (Φ2); 3 (Φ3); or L (Linker) (Fig. 3A). We used a combination of deletion and point mutants (annotated as 'Δ' and 'm' in Fig. 3A) together with quantitative measures of protein localization (partition coefficient) and dynamics (FRAP) to define the activity associated with each motif (complete data in Fig. EV3; Movies EV2–13). The partitioning parameter provides another key metric of protein activity by quantifying the relative distribution between heterochromatin compartments and the nucleoplasm. For reference, representative live-cell images are shown for each mutant protein together with the wild-type HRD (Fig. 3B). This indicated that most HRD mutants retained some level of chromocenter localization except for those targeting both the Φ2 and Φ3 motifs (Δ23 and m23 in Fig. 3B). Upon comparing partitioning data to recovery at 5 and 100 s using hierarchical clustering, four overarching groups were evident (Fig. 3C). These coincided with the individual contributions of CR1 (c4) and CR2 (c2), their cooperative activity (c3), and their combined deletion or mutation (c1). There was a clear trend in partitioning based on mutant topology, and, not unexpectedly, this inversely tracked with mobility (Fig. 3C, mobile fraction). Proteins with high partition coefficients showed low exchange with total chromocenter bleaching (WT in Fig. 3B), while those with no apparent heterochromatin localization (Δ23 and m23 in Fig. 3B) recovered almost instantaneously (Fig. EV3 and Movies EV2–13). As outlined below, these assays confirmed a bipartite HRD functional architecture and established that its activity reflected the complex interdependency of the two constituent modules (CR1 and CR2).

While CR1 (Δ3) was the main driver of dynamic behavior, retention was dependent on the presence of CR2 (Δ1HC2), even though it showed rapid mobility and only moderate chromocenter localization on its own (Figs. 3B,C and EV3). Moreover, the similarity between the CR2 deletion (Δ3) and its corresponding mutation (m3) indicated that the activity contributed by CR2 was largely through the Φ3 motif. This cluster also included the linker deletion (ΔL), suggesting there is a minimum length requirement between CR1 and the Φ3 motif. Further deletion of the C-terminal hydrophobic

motif (Φ2) from CR1 (Δ23) completely ablated heterochromatin localization and led to instantaneous recovery following photobleaching. The residual CR1 portion comprising the Φ1 and His-Cys motifs, therefore, lacks independent or appreciable binding determinants. This was corroborated by the corresponding double point mutant (m23) of the Φ2 and Φ3 motifs, suggesting these motifs constitute an essential chromocenter zip code. On the flip side, derivatives containing both the Φ2 and Φ3 motifs (Δ1HC) or only Φ3 (Δ1HC2) did not show substantive differences in mobility and clustered into the same subgroup, albeit the loss of Φ2 did reduce partitioning somewhat (c2.3 in Fig. 3C). Thus, the synergism contributed by Φ3 in chromocenter retention required the full CR1 domain. Single mutants targeting the Φ1, His, Cys, and Φ2 motifs (m1, mH, mC, and m2 in Fig. 3a) had similar effects on localization and mobility (Figs. 3B,C and EV3), showing they function in a coordinated manner that was consistent with the spatial constraint their sequence architecture was under. Each of these mutants does retain some level of heterochromatin localization and delayed recovery, which likely reflects residual contributions from Φ3 in conjunction with unmutated motifs in CR1. In this context, however, trends amongst deletions and point mutants of CR1 did show minor differences in initial recovery that gave rise to distinct subgroups within cluster 2 and together suggested a slightly greater dependence on Φ1 (c2.3 in Fig. 3C). The overall clustering of these CR1 mutants further reinforced that cooperativity achieved by tethering to Φ3 required the coordinated activity of all motifs (Φ1, His, Cys, and Φ2). Together, this indicates that CR1 uses an all-or-none mode of action that unmasks a latent and potent chromatin-binding feature.

## Chromocenter partitioning of the HRD shows "sink-like" behavior

We next carried out more detailed analyses of wild-type and mutant HRD partitioning between chromocenters and the nucleoplasm, which revealed another key insight into their respective localization behaviors. This leveraged the fact that partitioning can be evaluated over a range of protein expression levels to inform localization metrics as a function of concentration. Notably, while full-length KMT5C and the wild-type HRD exhibited a broad range of partition coefficients, it was more limited for the mutant and deletion derivatives (Fig. EV3). When individual partition coefficients were plotted as their normalized chromocenter (Y-axis) versus nucleoplasmic (X-axis) intensities, wild-type HRD (cluster 3 in Fig. 3C) showed progressive accumulation in chromocenters while maintaining a relatively constant (and minimal) nucleoplasmic level as expression increased (Fig. 4A, left panel, gray background). In contrast, the normalized intensity values for HRD mutants within chromocenters (cluster 2 in Fig. 3C) and the surrounding nucleoplasm leaned toward the diagonal over the same range of expression as inferred from total nuclear fluorescence intensity (Fig. 4A, center panel). In other words, the relationship between chromocenter and nucleoplasmic concentration remained relatively constant, which requires that the exchange rate is not varying appreciably as protein levels increase. As expected, mutants lacking chromocenter localization (cluster 1 in Fig. 3C) fell on the diagonal, reflecting similar intensities in both compartments (Fig. 4A, right panel). HRD mutants that retained a functional CR1 domain (cluster 4 in Fig. 3C), however, showed an intermediate phenotype (Fig. 4A, left panel). As for the FRAP

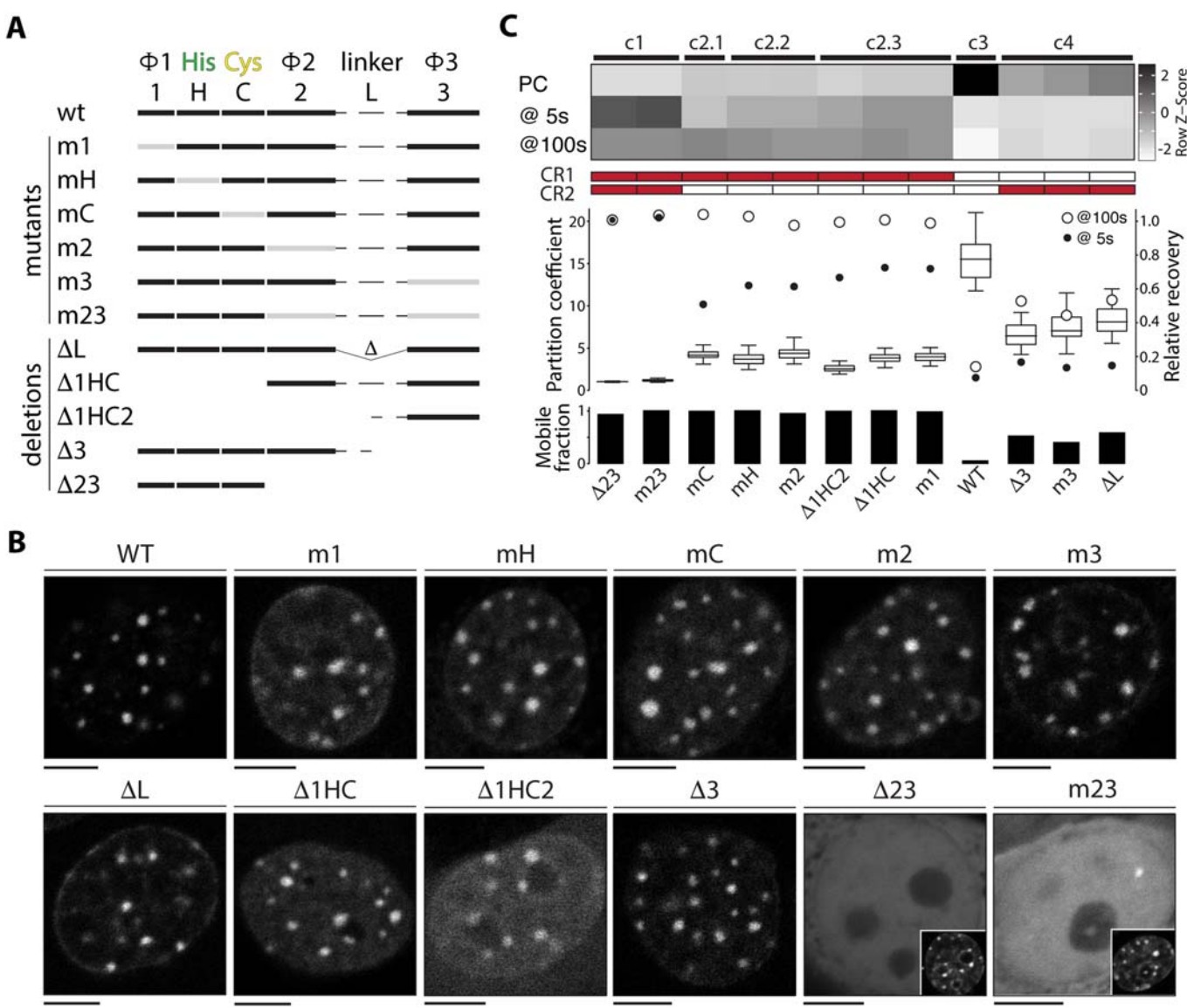

**Figure 3. Determinants of HRD localization and dynamics.**

(A) Schematic summary of HRD deletion (Δ) and point mutant (m) proteins. The Φ1, His, Cys, Φ2, linker, and Φ3 motifs are abbreviated as 1, H, C, 2, L, and 3. Point mutants are denoted by a light gray bar, while deletions are denoted by the absence of a bar. Each protein harbors a fluorescent fusion protein at its carboxyl terminus. (B) Representative live-cell images for the wild-type HRD and mutant proteins following transient transfection. The scale bar is 5 μm. For the Δ23 and m23 proteins, the inset includes the DAPI channel to demarcate the presence of chromocenters. (C) Heatmap depicting hierarchical clustering (c1, c2.1, c2.2, c2.3, c3, and c4) of HRD derivatives based on partition coefficient and fluorescence recovery at 5 and 100 s. Corresponding partition coefficient and FRAP recovery values are shown in graphs, together with the calculated mobile fraction from a double-term exponential curve fit. Filled boxes in the CR1 and CR2 rows summarize motif mutation status (red) within the HRD. For box plots, vertical lines indicate the bounds of the box and whiskers (minimum to maximum), and the box corresponds to the middle 50% of PC values (Q1–Q3), with the median indicated by a horizontal line. For FRAP and PC data, three separate experiments were conducted with a minimum of 15 cells each. Source data are available online for this figure.

analysis, this indicates that CR1 binding strength is the main driver of partitioning and that the added valency of CR2 augments preferential accumulation of the HRD over a wide expression range.

Together, the partitioning results also indicated that the relative concentrations of the HRD and intact CR1 derivatives (Δ3, m3, and ΔL) in chromocenters and the nucleoplasm are uncoupled to varying extents. This reflects an obvious bias towards influx (circular schematics in Fig. 4A), which dovetails with the iFRAP data that showed limited HRD efflux on long time scales (Fig. 1C)

and another contributing element to chromocenter retention over a broad concentration range. To test the limits of this behavior, we monitored HRD partitioning at the upper limits of exogenous expression (Fig. 4B, left). Even under these conditions, HRD chromocenter accumulation continued to increase over the full range of expression, although the rate of increase was attenuated across the highest nuclear fluorescence intensity values (Fig. 4B, left, inset). It was not until these maximal chromocenter levels were achieved (pseudo color in Fig. 4B, left, inset) that partition coefficient values decreased and plateaued just above 1 due to

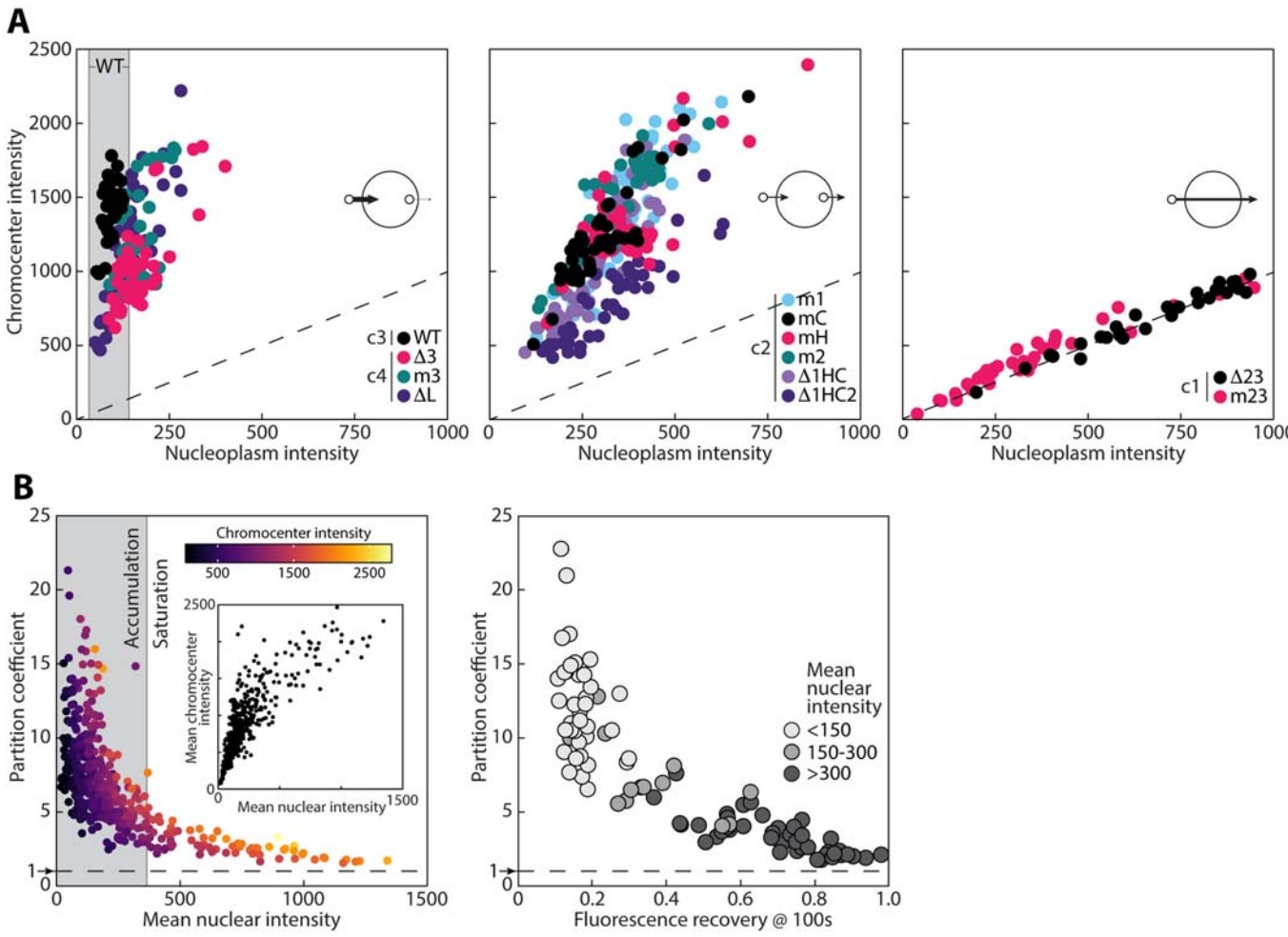

**Figure 4. Chromocenter partitioning behavior of wild-type and mutant HRD derivatives.**

(A) Scatter plots depict normalized intensities for the indicated proteins in chromocenters (Y-axis) versus nucleoplasm (X-axis). Left, data for clusters 3 (WT HRD in Fig. 3) and 4 (CR1-intact in Fig. 3) proteins; center, data for cluster 2 from Fig. 3; right, data from cluster 1 in Fig. 3. Dashed line indicates equivalent intensity values for chromocenter and nucleoplasm that would generate a partition coefficient value of 1. The schematic in each graph depicts a chromocenter with influx and efflux properties of each group of HRD derivatives shown as arrows with line weights representative of relative rates (proportionality not to scale). (B) Saturation analysis of wild-type HRD partitioning and exchange. Left, partition coefficient (Y-axis) plotted as a function mean HRD nuclear intensity value (X-axis) with respective chromocenter intensity shown as a gradient (MPL-inferno). Inset, the corresponding plot for mean intensity values for the HRD in chromocenters (Y-axis) versus nucleoplasm (X-axis). Right, HRD fluorescence recovery at 100 s (X-axis) as a function of partition coefficient (Y-axis), which is shaded as a function of mean nuclear intensity into low (mean nuclear intensity = 81, range = 34–138, n = 40), medium (mean nuclear intensity = 222, range = 151–284, n = 13), and high (mean nuclear intensity = 555, range = 309–1150, n = 42) categories. Source data are available online for this figure.

substantive HRD accumulation in the nucleoplasm. This decline in partitioning coincided with a progressive increase in HRD exchange in FRAP experiments (Fig. 4B, right), which approached full recovery at the highest protein levels, establishing that retention is concentration-dependent. These partitioning and FRAP profiles are consistent with saturation whereby HRD levels eventually exceed the number of available chromocenter interaction sites (the 'sink') and begin to 'spill-over' into the nucleoplasm.

## The HRD is reliant on HP1 interactions for chromocenter retention

The region comprising the HRD has been shown to physically interact with members of the Heterochromatin Protein-1 (HP1)

family (Souza et al, 2009; Hahn et al, 2013; Schotta et al, 2004; Bosch-Presegué et al, 2017), which comprises CBX1, 3, and 5 in mammals (Jones et al, 2000). Importantly, in these original studies, each half of the HRD could independently interact with HP1 members (Hahn et al, 2013), suggesting both segments contained an interaction determinant. In this context, HP1-interacting proteins typically contain a Pro-x-Val-x-Leu (PxVxL) pentapeptide motif (Lechner et al, 2005; Thiru et al, 2004; Smothers and Henikoff, 2000; Brasher et al, 2000) or a related hydrophobic sequence (Fig. 5A) (Liu et al, 2017). This motif is bound by a hydrophobic pocket created by HP1 dimerization through its chromo shadow domain (CSD), which makes additional contacts preceding and following the motif (Thiru et al, 2004; Richart et al, 2012). In this context, even though the Φ2 and Φ3 motif

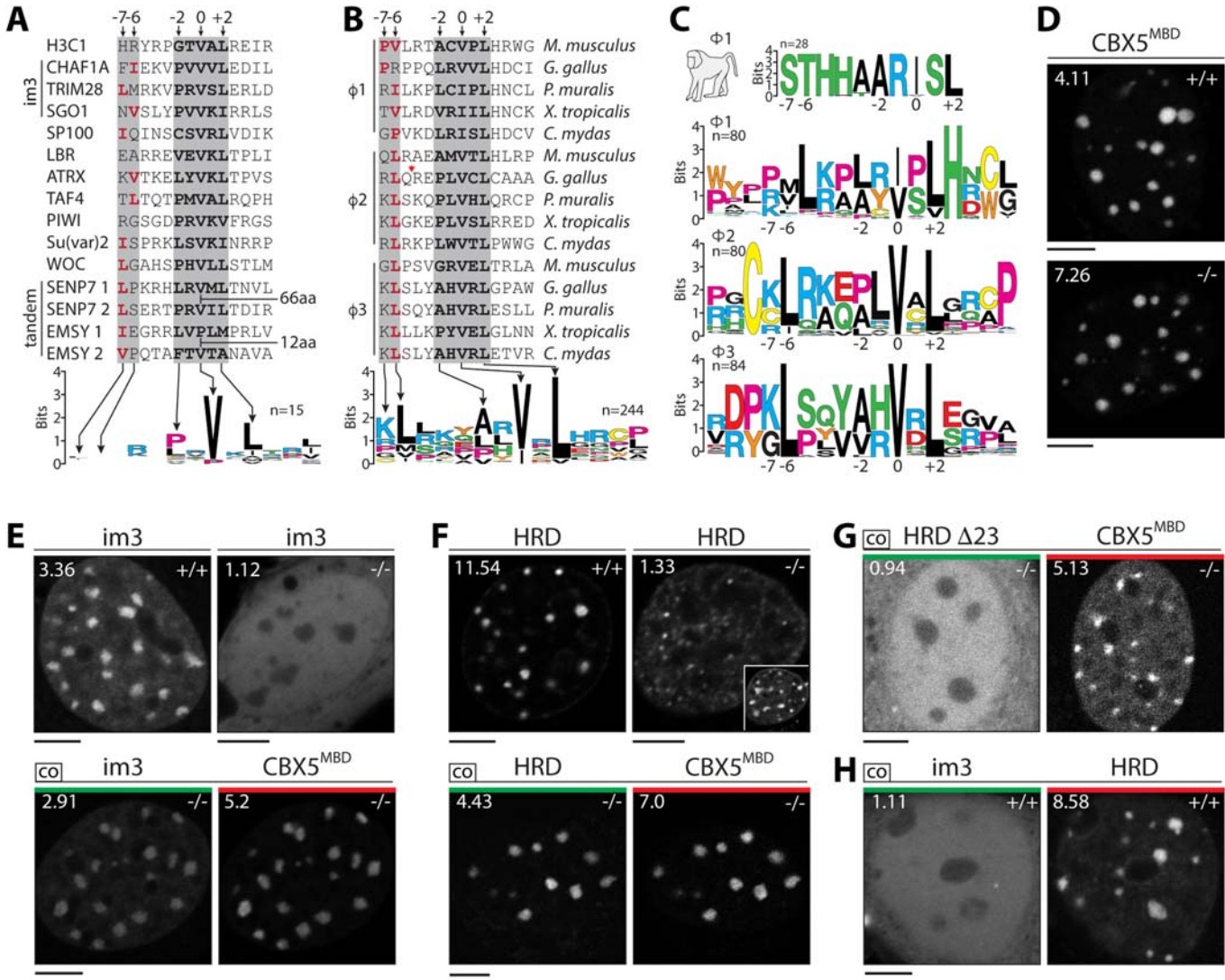

**Figure 5. Requirement for HP1 interactions for HRD localization and retention.**

(A) Alignment of known chromo shadow domain interaction motifs from the indicated proteins. The five-residue core motif and upstream residues are highlighted in gray, with the central position numbered as "0" and flanking residues assigned accordingly (+2, −2, −6, and −7). The presence of isoleucine, leucine, or valine residues at the −6 or −7 positions are shown with red lettering. The 3-motifs that make up the im3 synthetic protein are indicated. For SENP7 and EMSY, which contain tandem motifs, both are shown together with the center-to-center distance. The compiled WebLogo motif is shown below the alignment. (B) Alignment of representative hydrophobic motifs (Φ1, Φ2, and Φ3) from HRD orthologs in mammal (*Mus musculus*), bird (*Gallus gallus*), reptile (*Podarcis muralis*), frog (*Xenopus tropicalis*), and turtle (*Chelonia mydas*). Annotation is as described for panel (A), with the WebLogo compiled from 244 motif occurrences in HRD orthologs. For birds, the red arrow indicates the removal of one residue to facilitate alignment. (C) Individual WebLogo representation of Φ1, Φ2, and Φ3 motif classes (number of sequences indicated). For Φ1, the Primate version is shown separately to highlight its unique sequence composition compared to occurrences in other mammals, birds, reptiles, frogs, and turtles. (D) Representative images for the CBX5^MBD chimera in *Suv39h1/2* wild-type (+/+) and null (−/−) MEFs with average PC values indicated. (E) Representative images showing im3 localization in *Suv39h1/2* wild-type (+/+) and null (−/−) MEFs (upper panels) and upon CBX5^MBD co-expression (bottom panels; mEmerald or mCherry fusion proteins is denoted by a green or red bar); average PC values indicated on top left. (F) representative images for HRD localization in *Suv39h1/2* wild-type (+/+) and null (−/−) MEFs (upper panels) and upon CBX5^MBD co-expression (bottom panels, mEmerald or mCherry fusion proteins is denoted by a green or red bar); average PC values indicated on top left. (G) Representative images for co-expression of HRD Δ23 (mEmerald) and CBX5^MBD (mCherry) in *Suv39h1/2* null (−/−) MEFs shows dependence of rescue on the Φ2 and Φ3 motifs; average PC values indicated on top left. (H) Representative images for co-expression of the im3 (mEmerald) and HRD (mCherry) proteins in *Suv39h1/2* wild-type MEFs shows displacement of im3 by the HRD; average PC values indicated on top left. The scale bar is 5 µm. Images in panels (D–H) were derived from transiently transfected cells. For PC data, three separate experiments were conducted with a minimum of 15 cells each. Source data are available online for this figure.

architecture appeared atypical from a conservation standpoint, they do conform to other instances of the CSD binding motif (including the canonical PxVxL version), and the first leucine (-6 position) lies in the region of extended contact with the CSD (Fig. 5B,C). While Φ1 largely adhered to the degenerate CSD-interacting motif

(Liu et al, 2017) by virtue of having hydrophobic residues at the −2, 0, and +2 positions, the 0 position was frequently occupied by isoleucine, and it showed greater variability overall, including the -6 and -7 locations (Fig. 5C). This was especially notable in Primates, which lacked hydrophobic residues at the upstream positions,

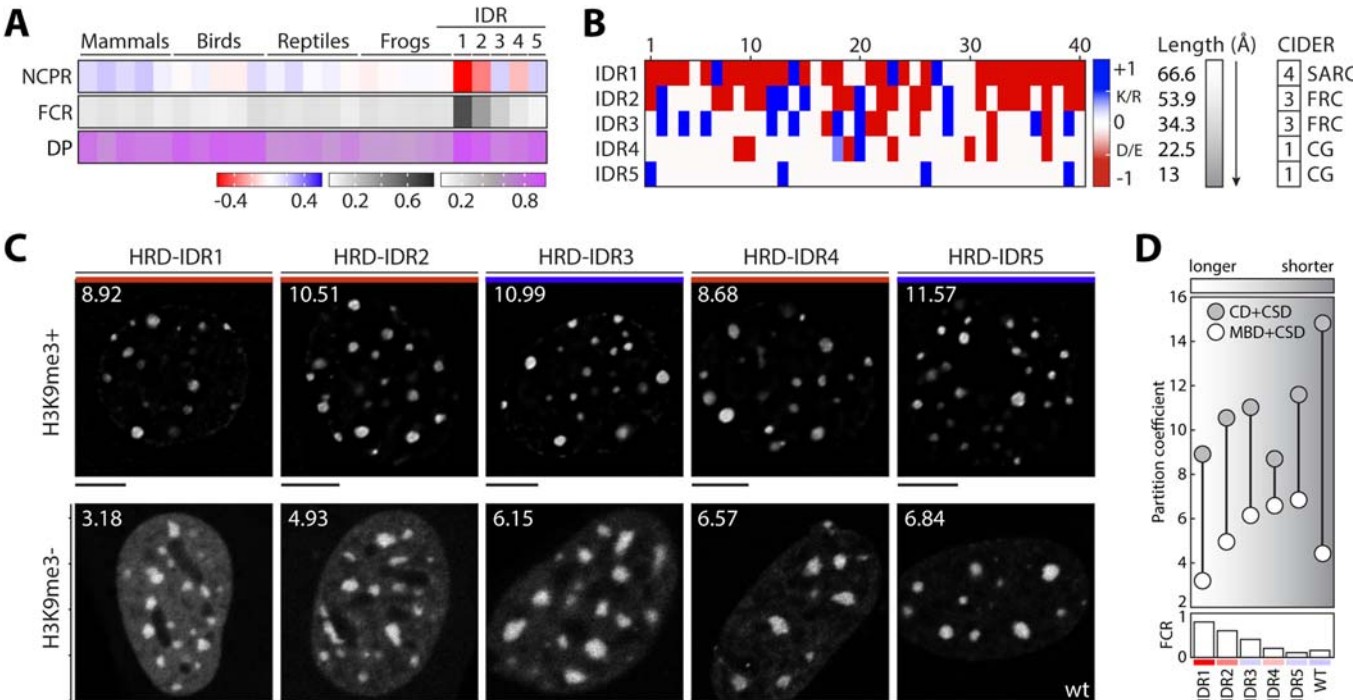

**Figure 6. Contributions of the linker region to HRD chromocenter retention.**

(A) Sequence features of HRD natural and chimeric linkers. Five representative linker sequences from the indicated species classes (Mammal, Bird, Reptile, and Frog) are shown in comparison to the five chimeric sequences (IDR1-5). For each, heatmaps show net charge per residue (NCPR), fraction of charged residues (FCR), and fraction of disorder-promoting (DP) amino acid residues. Corresponding intensity scales for each feature are shown below the heatmap. (B) EMBOSS charge profiles are shown for the 40-residue chimeric IDR linker series to illustrate the distribution of positive (K and R in blue) and negative (D and E in red) amino acids (intensity scale shown on right). Based on increasing FCR and CIDER analysis, linkers are assigned to categories that include compact globules (CG, category 1), Flory random coils (FRC, category 3), or self-avoiding random coils (SARC, category 4). (C) Representative images for HRD linker chimeras in NMuMG cells (top row) or upon co-transfection with CBX5[MBD] in *Suv39h1/2* null cells (bottom row). The red and blue bars distinguish images corresponding to linkers with either negative or positive NCPR values. Images were derived from transiently transfected cells. The scale bar is 5 μm. Average PC values are indicated on the top left of each image. (D) Summary of average PC values for the HRD and IDR chimeras in *Suv39h1/2* wild-type (CD-CSD and H3K9me3+) or null (MBD-CSD and H3K9me3−) cells. Differences in linker functional length are shown as a gradient (from left to right), the magnitude of NCPR for each linker is shown in panel (A), and FCR is plotted below with a separate scale. For PC data, three separate experiments were conducted with a minimum of 15 cells each. Source data are available online for this figure.

instead having an invariant threonine-histidine pair (Fig. 5C, top). Moreover, mutagenesis indicated that Φ1 does not function as a standalone heterochromatin localization motif in the context of the Δ23 deletion or m23 mutant (Fig. 3B).

Given the redundancy between CBX1, 3, and 5, and the challenge it created for a loss-of-function, we opted to use two orthogonal approaches to evaluate the contribution of Φ motifs to HP1-dependent localization relying on chromocenter partitioning as a functional readout (full data shown in Fig. EV4). First, we leveraged the fact that the MeCP2 methyl-DNA binding domain (MBD) retains chromocenter localization (Cooper et al, 2014) in *Suv39h1/2* null cells (Peters et al, 2001), which lack chromocenter H3K9me3 and do not support HP1 recruitment (Lachner et al, 2001). We therefore created a chimeric CBX5 protein in which the chromodomain was replaced by the MBD (CBX5[MBD]). This protein showed robust chromocenter localization in both *Suv39h1/2* wild-type and null cells with partition coefficients of 4.11 and 7.26, respectively (Fig. 5D), allowing us to assess CSD-dependent recruitment. As proof-of-principle, we employed a synthetic protein containing 3 well-characterized CSD interaction motifs (Fig. 5A) that supported chromocenter localization in *Suv39h1/2*

wild-type cells, but not in the null setting (Fig. 5E, top). Upon CBX5[MBD] co-expression, however, partitioning of this protein increased from 1.12 to 2.91 in *Suv39h1/2* null cells (Fig. 5E, bottom), which approached the value of 3.36 in wild-type cells (Fig. 5E, top). Likewise, while the partition coefficient of the HRD decreased from 11.54 in wild-type *Suv39h1/2* cells to 1.33 in null cells (Fig. 5F, top), the CBX5[MBD] chimera was able to restore localization (Fig. 5F, bottom; partition coefficient increased to 4.43). In line with the FRAP and partitioning data (Fig. 3C), rescue was dependent on the Φ2 and Φ3 motifs (Fig. 5G, Δ23 shown). This indicated that these motifs are the main drivers of CSD interactions and heterochromatin localization, and that the Φ1 motif (retained in Δ23) had negligible activity in this context. Second, using an "in cell" competition assay, the HRD could fully displace the synthetic CSD-interacting protein from chromocenters in wild-type cells with an intact SUV39H1/2 pathway (Fig. 5H; partition coefficient value decreased from 3.36 to 1.11). Consistent with competing for a common pool of endogenous HP1, this coincided with a reduction in HRD partitioning from 11.54 to 8.58 (Fig. 5H). Combined with the established HRD-HP1 physical interactions (Souza et al, 2009; Hahn et al, 2013; Schotta et al, 2004;

Bosch-Presegué et al, 2017) and the relatedness of Φ2 and Φ3 to known CSD interaction motifs (Fig. 5A,B), these cell-based results provide further evidence for a prominent role of HP1 proteins in HRD localization.

## Optimal tethering requires a compact globular linker

The data thus far support an essential role for tethering CR1 and CR2 to convey HRD retention, which prompted a more detailed characterization of the intervening linker sequence. Although not conserved in terms of alignable sequence (Fig. 2A), it was nevertheless under evolutionary constraints for other attributes. In general, the entire region carboxy-terminal to the structured SET methyltransferase domain of KMT5C orthologs was intrinsically disordered (Emenecker et al, 2021; Tesei et al, 2024), including the HRD (Fig. EV5). This was particularly evident for the linker region, which also consistently showed a low fraction of charged residues (FCR) (Das and Pappu, 2013) and a small net charge per residue (NCPR) (Das and Pappu, 2013) value that was almost invariably positive (Fig. 6A; Datasets EV1 and 2). These physicochemical properties would suggest the formation of a compact globule that would shorten the functional length between Φ2 and Φ3 (Mao et al, 2010; Holehouse et al, 2017). To explicitly query the role of these features in the linker function, we used a series of well-defined intrinsically disordered sequences of a fixed length that encompassed different conformational ensembles and charge properties (Harmon et al, 2017). At 40 residues, these linkers were longer than the endogenous version in mammalian and bird HRDs, but shorter than those in reptiles and frogs. They included linkers that act as self-avoiding random coils (SARCs), Flory random coils (FRCs), and compact globules as FCR decreases (Harmon et al, 2017), leading to progressively shorter functional lengths of 66.6, 53.9, 34.3, 22.5, and 13 Å (Harmon et al, 2017) (Fig. 6B).

This analysis revealed two important facets of HRD activity. First, the linker replacement chimeras largely retained the retentive behavior in FRAP experiments (Fig. EV6A; Movies EV14–18), indicating the two modules are the principal drivers of kinetic behavior. Each chimera showed a slight increase in exchange rate but there was no obvious relationship with linker type (FCR or NCPR) or composition (Fig. EV6A). There was, however, a general decrease in partition coefficients (Fig. 6C, upper panels) with the transition from compact globule to SARC linkers and an obvious charge influence (Figs. 6D and EV6). Specifically, IDRs with positive NCPR partitioned more favorably than their negative counterparts, and negative charge was particularly detrimental in more compact linkers, as evident with the IDR4 chimera (Fig. 6C,D). We also evaluated the IDR chimera series using the CBX5$^{MBD}$ rescue strategy in *Suv39h1/2* null cells (Fig. 6C, lower panels) to determine if the differences solely reflected CSD-mediated effects or had a more complex etiology. In this setting, the IDR series showed a progressive increase in partitioning that correlated ($R^2 = 0.98$) with the decreasing functional length of the linker regardless of the NCPR value (Fig. 6C,D), which would be consistent with influencing the effective concentration of motif interactions (Dyla and Kjaergaard, 2020). Moreover, the IDR2-IDR5 chimeras achieved partitioning that exceeded that of the wild-type HRD, indicating a higher level of binding efficiency with the CBX5$^{MBD}$ protein (Fig. 6C,D). This suggested that the endogenous HRD linker was not optimized for CSD-driven cooperativity

between the Φ2 and Φ3 but was well-adapted for localization in H3K9me3-competent cells where it exhibited the highest partition coefficient (Fig. 6C,D). As such, linker activity is responsive to (or can sense) the heterochromatin environment. Thus, comparing the same chimeric series of proteins in two different epigenetic settings revealed additional layers of complexity that indicated the HRD linker may be under functional constraints applied by the entire CR1 module and not simply for CSD binding.

## Discussion

Protein dynamics in constitutive heterochromatin compartments have been studied for over 20 years, starting with HP1 (Cheutin et al, 2003; Straub, 2003; Maison and Almouzni, 2004), and continue to provide surprises. This includes FRAP studies of HP1α (CBX5), HP1β (CBX1), HP1γ (CBX3) (Cheutin et al, 2003; Muller-Ott et al, 2014), SUV39H1/2 (Krouwels et al, 2005; Muller-Ott et al, 2014), and KMT5B/C (Souza et al, 2009; Hahn et al, 2013; Muller-Ott et al, 2014), which together form the core epigenetic framework of constitutive heterochromatin (Saksouk et al, 2015; Schotta et al, 2004). Collectively, these proteins defined a continuum of dynamic behaviors when assessed for recovery following total photobleaching of chromocenters: all 3 HP1 proteins exhibited rapid exchange, SUV39H2 and KMT5C showed very limited exchange and were thought to be immobile, while SUV39H1 and KMT5B were intermediate. Despite these enduring views, we show that lack of exchange is not necessarily reflective of lack of mobility or stable binding. To the contrary, using a partial chromocenter photobleaching strategy, we observe that equilibration of fluorescence internally within these heterochromatin compartments can occur more than two orders of magnitude faster than from nucleoplasmic exchange. At the same time, we know from our previous work that the chromatin itself forms an immobile scaffold (Strickfaden et al, 2020). The behavior of KMT5C is especially notable because it relies on a protein, HP1, that is otherwise highly dynamic, and because it is achieved using limited sequence features. Mobility of KMT5C within chromocenters also argues against a recently reported role for "locking" HP1 on constitutive heterochromatin (Nakao et al, 2024).

HP1 protein interactions are described primarily in the context of CSD binding to canonical PxVxL-type motifs (Lechner et al, 2005; Thiru et al, 2004; Smothers and Henikoff, 2000). While a largescale screen supports the prevalence of this binding mode, alternative interaction types exist (Nozawa et al, 2010). Notably, the POGZ protein interacted with the CSD using a Cys$_2$-His$_2$ zinc finger-like motif in the absence of hydrophobic motifs, and the same sequence was required for heterochromatin localization (Nozawa et al, 2010). Although the HRD His-Cys feature may also constitute a zinc finger-like motif (see below), this was unrelated to the POGZ version in sequence and architecture and showed no propensity for heterochromatin localization on its own (Δ23 in Fig. 3B). Using phylogenetic annotation (Figs. 2A and 5B,C) and functional analyses (Figs. 3B,C and 5D–H), we show instead that localization was dependent on dual hydrophobic motifs (Φ2 and Φ3). These appear to constitute a unique subclass of HP1-interaction motifs not previously described in KMT5C sequence annotation (Nakao et al, 2024) and clearly different from Φ1 in sequence and activity. Like the HRD, both the EMSY (Huang et al,

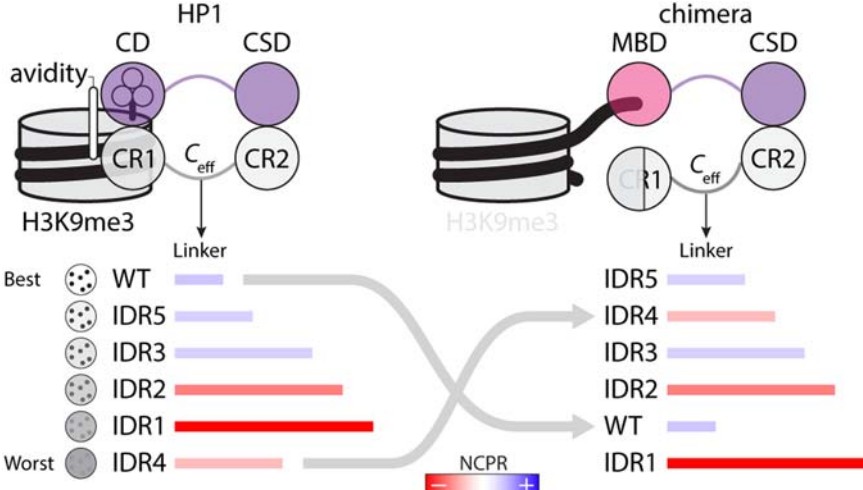

**Figure 7. Mechanistic overview of drivers of HRD retention and partitioning.**

Cooperativity in wild-type cells (left panel) involves both the chromodomain (CD) and chromo shadow domain (CSD) of HP1 (left). This is modeled as an avidity-driven process involving CD binding of H3K9me3 (lollipop) and additional nucleosome engagement by CR1 to increase affinity. Tethering to CR2 facilitates HP1 effective concentration ($C_{eff}$) to confer elevated affinity and retention. The linker influence on cooperativity is shown as a function of partitioning (decreasing from top to bottom) with linkers colored according to NCPR (as in Fig. 6). On the right, the CBX5$^{MBD}$ binding mode is shown, which involves DNA engagement by the MBD. In the absence of H3K9me3 and CD involvement, the Φ1, H, and C motifs appear to make minimal contributions (depicted as a partial mask over CR1), such that increased effective concentration is acquired largely through CSD interactions with Φ2 and CR2 (Φ3). For this mode, the wild-type HRD shows inferior partitioning in comparison to the IDR2-5 chimeras, whose partitioning overall is highly correlated to functional length.

2006; Ekblad et al, 2005) and SENP7 (Maison et al, 2012; Romeo et al, 2015) proteins use a strategy where cooperativity between tandem hydrophobic motifs confers efficient CSD-dependent chromocenter localization. Additional hydrophobic contacts with the CSD flanking the core motif, as may be the case for Φ2 and Φ3 (Fig. 5B,C), can increase the affinity of HP1 interactions (Thiru et al, 2004; Mendez et al, 2013; Liu et al, 2017). Nevertheless, the HRD represents a further innovation in HP1-mediated targeting through the addition of adjacent motifs (Φ1 and His-Cys) that dramatically increased heterochromatin partitioning and enabled retention. In particular, Φ2 was necessary to unlock the latent binding activity of the His-Cys motif, potentially operating as a switch-like "AND" logic gate (Chen et al, 2020) in response to HP1 interaction. That these motifs were under spatial constraint suggests they function through avidity (Holehouse and Kragelund, 2024), where they could extend the HP1-chromatin interface, potentially by the formation of a zinc finger that leads to a considerable affinity boost (Fig. 7). Together with the rapid evolution and species specificity of satellite DNA (Plohl et al, 2008; Mestrovic et al, 2015; Brand and Levine, 2021; Charlesworth et al, 1994), this would fit with the varied spacing between the histidine and cysteine residues in CR1 and overall divergence amongst HRD orthologs.

As well adapted as CR1 was for efficient heterochromatin localization and only moderate nucleoplasmic exchange, its tethering to an additional hydrophobic motif (Φ3) in CR2 further enhanced partitioning, while greatly suppressing exchange. Although tethering was achieved using a highly variable linker that was not under the same spacing constraints as CR1 motifs, it was invariably characterized by low FCR that was strongly biased against negative NCPR values. The importance of these features was confirmed using a series of IDR linker chimeras with defined

charge properties and sequence ensemble relationships (Harmon et al, 2017). Together, they were consistent with the requirement of a compact globular linker with a short functional length (Das and Pappu, 2013; Mao et al, 2010; Müller-Späth et al, 2010; Harmon et al, 2017). This was particularly evident with the CBX5$^{MBD}$ rescue of the HRD-IDR series, which showed the strongest negative correlation between functional length and partitioning (Figs. 6 and 7). A striking example of such 'conformational buffering' is provided by the adenoviral E1A protein (González-Foutel et al, 2022). Like the HRD, two short linear motifs in E1A are tethered by a hypervariable linker whose functional length is under evolutionary constraint imposed by amino acid composition. This enables strong binding to the retinoblastoma protein through an effective concentration mechanism that enhances affinity 4000-fold, allowing it to out-compete cellular host proteins. In other disordered tether paradigms, this affinity increase is driven largely by a reduction in $k_{off}$ values (Dyla et al, 2022). By analogy, tethering of CR1 and CR2 may drive higher affinity such that it becomes a "preferred client" of HP1, an idea borne out by the ability of the HRD to completely displace the synthetic HP1-interacting protein from chromocenters (Fig. 5H). This strategy, where partitioning gives a robust quantitative readout, can, therefore, serve as a model system for how effective concentration driven by intrinsically disordered linkers (Dyla and Kjaergaard, 2020; Sorensen and Kjaergaard, 2019) controls protein localization in complex cellular environments.

Differences in the partitioning profiles of the HRD and HRD-IDR chimeras in the *Suv39h1/2* wild-type and null settings indicated the linker also played critical and context-dependent roles in modulating CR1 and CR2 cooperativity (Fig. 7). First, while partitioning of the chimeras, in general, scaled proportionally with functional length in both cell backgrounds, the presence of negative

NCPR in IDR4 did incur a penalty in the wild-type setting. Second, and more striking, the HRD showed a disproportionate decline in partitioning in the *Suv39h1/2* null condition despite being the most efficient in wild-type cells (Fig. 6). To account for this, we suggest the HRD can adopt high (Fig. 7, left) and low (Fig. 7, right) affinity states in response to chromocenter H3K9me3 status and HP1 chromodomain involvement. The endogenous HRD linker is either not compatible with the low-affinity state or may actively suppress it, consistent with the potential for intrinsic disorder to provoke autoinhibition (Fenton et al, 2023). Notably, this barrier did not exist for the chimeric linkers, establishing that the activity resides within the endogenous linker. In the presence of H3K9me3, however, the endogenous linker was best suited for the high-affinity binding state achieved using CR1-CD avidity (Fig. 7, left). We speculate the linker may enforce the high-affinity state by "sensing" H3K9me3 status, adding to the idea that intrinsically disordered regions can respond to their environment to modulate protein function (Moses et al, 2023; Moses et al, 2024). Collectively, the stoichiometry and spatial density of HP1 and H3K9me3 within heterochromatin compartments will determine the balance between affinity states with the linker providing an added specificity determinant.

HP1 has been proposed to undergo liquid-liquid phase separation in association with heterochromatin (Larson et al, 2017; Strom et al, 2017), which occurs upon exceeding a saturation concentration to produce a 2-phase system: dense and dilute. Nevertheless, alternative methods of HP1 compartmentalization have been proposed (Erdel et al, 2020; Muzzopappa et al, 2022). While preferential internal mixing of KMT5C and the HRD fit key criteria for phase separation (Muzzopappa et al, 2022), the sustained efficiency of HRD chromocenter partitioning over a wide concentration range and its saturability suggest otherwise. We propose this behavior reflects the nature of the binding event and the availability of binding sites. Specifically, efficient binding is achieved through a combination of CR1 avidity and maintenance of a high local HP1 concentration by CR2, together with an abundance of H3K9me3 binding sites that are spatially confined on an immobile, gel-like chromatin scaffold (Belaghzal et al, 2021; Strickfaden et al, 2020) (Fig. 7). This combination of features may facilitate KMT5C 'hopping' or transfer between chromatin-binding sites, possibly via facilitated dissociation (Erbaş and Marko, 2019). The bipartite nature of the HRD, with CR1 and CR2 constituting distinct affinity states, could add to this phenomenon. Moreover, exceeding the number of chromatin-binding sites is unlikely to occur at endogenous KMT5C protein levels. Notably, querying quantitative proteomics data from tissues (Wang et al, 2019) and breast cancer cell lines (Lawrence et al, 2015) indicate KMT5C levels are at least 2–3 orders of magnitude lower than HP1 proteins. The vast difference in stoichiometry should further contribute to KMT5C chromatin binding/rebinding by ensuring HP1 is not in limited supply. In this regard, previous analyses of an endogenously GFP-tagged *Kmt5c* allele documented the same lack of nucleoplasmic exchange we observed, which provides an important benchmark even though internal mobility was not assessed (Hahn et al, 2013). That this exchange deficit persists upon exogenous KMT5C expression, supports the presence of considerable buffering capacity. Last, although the exogenous expression strategy used here could be considered a limitation, it was well suited to demonstrate the extremes of KMT5C chromocenter retention and the elegant strategy employed by the HRD to achieve this.

# Methods

## Reagents and tools table

| Reagent/resource | Reference or source | Identifier or catalog number |
|---|---|---|
| **Experimental models** | | |
| NMuMG cell line | Ing Swie Goping Lab | CRL-1636 |
| Suv39h1/2 +/+ (W8) | Thomas Jenuwein Lab | N/A |
| Suv39h1/2 −/− (D5) | Thomas Jenuwein Lab | N/A |
| **Recombinant DNA** | | |
| All plasmids (e.g., KMT5C-mEmerald) | Biomatik | Dataset EV3 |
| **Chemicals, enzymes and other reagents** | | |
| Transfection reagent | Qiagen Effectene Transfection Reagent | Cat. No. / ID: 301425 |
| DMEM | Gibco | 12800-082 |
| Fetal bovine serum (FBS) | Gibco | 12483-020 |
| Hoechst | Invitrogen | H3570 |
| **Software** | | |
| Prism - Graphpad | https://www.graphpad.com/features | N/A |
| ImageJ | https://imagej.net/ij/ | N/A |
| Zeiss LSM 5 ZEN Black | https://www.zeiss.com/microscopy/en/products/software/zeiss-zen.html | N/A |
| Microsoft Excel | https://www.microsoft.com/en-ca/microsoft-365/excel | N/A |
| Volocity® 6.3 software | https://www.volocity4d.com/ | N/A |
| **Other** | | |
| 35 mm dish \| No. 1.5 coverslip \| 14 mm glass diameter \| Uncoated | Mattek | P35G-1.5-14-C |
| Confocal microscopes | Cross Cancer Institute Cell Imaging Facility | N/A |

## Material availability

Materials developed in this study are available upon request but may require an MTA if intended for commercial use.

## Cell culture and transfection

Cells were cultured at 37 °C and 5% $CO_2$ in a humidified incubator. All cell lines were grown in DMEM containing 10% FBS. D5 (*Suv39h1/2* knockout) and W8 (*Suv39h1/2* wild-type) mouse embryonic fibroblast cell lines (Peters et al, 2001) were obtained from Dr. Thomas Jenuwein. All other analyses were carried out using the mouse NMuMG immortalized breast epithelial cell line (Owens et al, 1974). Cells were transfected by lipofection using Effectene (Qiagen) 1 day prior to experiments. Expression plasmids (including all deletion and point mutants) were synthesized (www.biomatik.com), obtained from the Addgene repository (www.addgene.org), or previously described (Tsang et al, 2010). For recombinant proteins developed for this study, corresponding sequences are provided in Dataset EV3. All plasmids

were sequence verified. Either mEmerald or mCherry tags were used for all recombinant fluorescent fusion proteins. For point mutations, the most conservative substitutions possible were made to minimize changes in physicochemical properties and confounding phenotypes.

## Live-cell imaging

Live-cell imaging was carried out using Zeiss Axiovert 200 M inverted microscopes attached to either an LSM510 NLO laser scanning system with a 25 mW argon laser line, a Zeiss LSM 770 confocal microscope attached to an Axio Observer Z3 equipped with 405, 488, 561, and 633 nm diode lasers, or a PerkinElmer UltraVIEW® spinning-disk confocal microscope equipped with 405, 488, and 561 nm diode lasers and a FRAP-unit. For all platforms, a 40× 1.3 NA oil immersion lens was used. Long-term live-cell observations were conducted on the spinning-disk microscope at 37 °C with humidification and 5% $CO_2$. Fluorescence recovery after photobleaching was performed on transiently transfected cells using the 488 nm solid state (spinning-disk confocal) or 488 nm argon laser line (LSM510). Circular regions were demarcated on top of (total chromocenter bleach) or adjacent to (partial chromocenter bleach) chromocenters and subsequently photobleached by intense light from the 488 nm laser. Fluorescence recovery of the bleached regions was quantified over multiple time scales (seconds to minutes). FRAP data was extracted using Zeiss LSM 5 ZEN Black or ImageJ software (Schneider et al, 2012) by measuring the fluorescence intensity of the background, the whole nucleus, and the bleached area in each of the recorded time-lapse pictures for a minimum of 45 cells (three independent experiments with a minimum of 15 cells each). Relative intensity (including standard deviation) was background corrected and normalized to account for photobleaching from scanning in Microsoft Excel and plotted using GraphPad Prism software. Additional FRAP parameters were extracted using the EasyFRAP web-based tool (Koulouras et al, 2018). For inverse FRAP (iFRAP), a mask that covered all chromocenters except one was used, and the entire cell was considered as the bleached region and corrected using the background outside of the cell. Fluorescence of the region of interest was normalized to fluorescence of the whole cell to account for photobleaching from scanning.

## Cell culture, transfection, and imaging experiments

1. Seed Mattek dishes 1 day before imaging with enough cells to achieve 80% confluency in 24 h.
2. Reverse transfect with Effectene (Qiagen) transfection reagent and 300 ng of plasmid DNA.
3. Incubate at 37 °C and 5% $CO_2$ in a humidified incubator overnight.
4. The next day, transfer Mattek dish to humidified chamber at 37 °C and 5% $CO_2$ on relevant microscope.
5. For FRAP imaging, demarcate circular regions on top of or adjacent to chromocenters and subsequently photobleach using intense laser light. Measure fluorescence intensity of the total and partially bleached chromocenters, the entire cell nucleus, and a background region (four ROIs) by imaging every 1 s post-bleach for 5 s and then every 5 s for another 100 s. Save the ROI intensity table and microscope image for further analysis. Repeat on a minimum of 15 cells for three independent replicates.
6. For PC imaging, take 2D images of transfected cells and save for further analysis. Repeat on a minimum of 15 cells for three independent replicates.

7. For all imaging of a single experiment, use the same microscope and identical imaging parameters.

## Image analysis

Kymographs were acquired for regions of interest from FRAP time series using the ImageJ Multi Kymograph tool. In both FRAP and partition coefficient analyses, laser power settings were initially optimized for cells with low levels of HRD expression to establish physiologically relevant baseline behaviors. For saturation analyses, the laser power was decreased to enable fluorescence intensity measurements in cells with higher expression of the HRD protein and its derivatives. These settings corresponded to a 1.3–1.7-fold decrease in intensity values compared to those over a low expression regime. To extract partition coefficient data, confocal microscopy images of a minimum of 45 transfected live-cell nuclei (three independent experiments with a minimum of 15 cells each) were imported into ImageJ and cropped to generate 2D images of single nuclei. The Trainable Weka Segmentation 3D plugin (Arganda-Carreras et al, 2017) was used to generate intensity-based masks of chromocenters, nucleoplasm, and background. These masks were applied to the original image to calculate the average intensity measurements associated with each. The background was subtracted from chromocenter and nucleoplasm intensities. Partition coefficients were then calculated by dividing corrected average chromocenter intensities by corrected average nucleoplasm intensities for each cell nucleus. Hoechst staining was used to generate masks for proteins with minimal partitioning to chromocenters.

## Fluorescence recovery after photobleaching (FRAP) data analysis

1. Background correction: Subtract the background intensity from the average intensity of each ROI (totally bleached chromocenter, partially bleached chromocenter, whole nucleus).
2. Use the following equation for normalization to correct for photobleaching from scanning, where $t_1$ is the first timepoint post-bleach, ROI1 is the bleached region (totally bleached or partially bleached chromocenter), and ROI2 is the whole nucleus.

$$Norm\, I(t_n)_{ROI1} = \frac{I(t_1)_{ROI2}}{I(t_n)_{ROI2}} * I(t_n)_{ROI1}$$

3. Calculate relative intensity at each timepoint using the following equation, where $t_1$ is the pre-bleach timepoint.

$$Relative\, Fluoresence\,(t_n) = \frac{NormI(t_n)}{I(t_1)}$$

4. Calculate the average relative fluorescence and standard deviation (SD) at each timepoint. Plot average relative fluorescence (background corrected and normalized) vs time with SD error bars.

## Partition coefficient (PC) data analysis

1. Open the PC image on ImageJ.
2. Open the Trainable Weka Segmentation 3D plugin.
3. Create a "chromocenter," "nucleoplasm," and "background" class.

4. Train the classifier by drawing two to three lines within each respective class.
5. View the overlay to determine if classification is acceptable. After successful classification, acquire probability masks.
6. Apply the probability masks to the original image to measure the average intensities of the "chromocenter," "nucleoplasm," and "background" ROIs.
7. Save intensity values as an .xls file
   a. Repeat with all PC images.
   b. For proteins that are not enriched at chromocenters, use the corresponding Hoechst image to train the classifier and create masks for the chromocenter, nucleoplasm, and background regions of the nucleus.
   c. An ImageJ macro can be written to automate the process.
8. Calculate the partition coefficients of each individual cell using the following formula:

$$\text{Partition Coefficient} = \frac{\text{Average Chromocenter Intensity} - \text{Average Background Intensity}}{\text{Average Nucleoplasm Intensity} - \text{Average Background Intensity}}$$

## Sequence analyses

In addition to the recovery of all tetrapod KMT5C orthologs annotated in the NCBI database (https://www.ncbi.nlm.nih.gov/gene/84787/ortholog/?scope=32523&term=KMT5C) and by Grau-Bové et al (Grau-Bové et al, 2022), the Basic Local Alignment Search Tool (BLAST) (Altschul et al, 1990) was used to expand this list to include unannotated orthologs in birds and reptiles using tblastn and a range of genome and transcript level sequence databases. Multiple alignments were carried out independently for CR1 and CR2 using ClustalΩ (Sievers et al, 2011) and sequence conservation was displayed using WebLogo (Crooks et al, 2004). Disorder profiles were derived using Metapredict (Emenecker et al, 2022) (https://metapredict.net/#). Per residue charge properties were determined using EMBOSS (https://www.bioinformatics.nl/cgi-bin/emboss/charge) with a window size of 1 and rendered as heatmaps using Heatmapper (http://www1.heatmapper.ca/) (Babicki et al, 2016). Protein sequence features were obtained using the ProtPram tool (https://web.expasy.org/protparam/) (Gasteiger et al, 2005) and LocalCIDER (Das and Pappu, 2013).

## Statistical analysis

For partition coefficients, significance was evaluated using the Kruskal–Wallis one-way analysis test and individual comparisons between proteins were done using the Wilcoxon rank-sum test.

# Data availability

No data used in this study was submitted to public databases.

The source data of this paper are collected in the following database record: biostudies:S-SCDT-10_1038-S44319-024-00320-5.

# Peer review information

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

## Acknowledgements

The authors acknowledge funding support from the Canadian Breast Cancer Foundation (DAU, grant no. 300073), Cancer Research Society (DAU and MJH, grant no. CRSDI 2018 OG 23446), and Canadian Institutes of Health Research (MJH, grant no. PJT-148753). JWK acknowledges studentship support from multiple sources (William Herbert Young Cancer Research Graduate Studentship sponsored by the Cancer Research Institute of Northern Alberta and Alberta Cancer Foundation, with additional funding from the Faculty of Medicine & Dentistry (University of Alberta) and Government of Alberta). The authors thank Dr. Xuejun Sun, Gerry Barron, and Dr. Guobin Sun of the Cross Cancer Institute Cell Imaging Facility for their support. DAU thanks the Fragile Nucleosome, IDP Seminars, and Condensate Colloquium Series for the many presentations that aided in creating this narrative.

## Author contributions

**Justin W Knechtel**: Data curation; Formal analysis; Investigation; Visualization; Methodology; Writing—review and editing. **Hilmar Strickfaden**: Data curation; Formal analysis; Investigation; Visualization; Methodology; Writing—review and editing. **Kristal Missiaen**: Data curation; Formal analysis; Investigation; Visualization; Methodology. **Joanne D Hadfield**: Data curation; Investigation; Visualization; Methodology. **Michael J Hendzel**: Conceptualization; Supervision; Funding acquisition; Methodology; Project administration; Writing—review and editing. **D Alan Underhill**: Conceptualization; Supervision; Funding acquisition; Visualization; Methodology; Writing—original draft; Project administration; Writing—review and editing.

Source data underlying figure panels in this paper may have individual authorship assigned. Where available, figure panel/source data authorship is listed in the following database record: biostudies:S-SCDT-10_1038-S44319-024-00320-5.

## Disclosure and competing interests statement

The authors declare no competing interests.

# Expanded View Figures

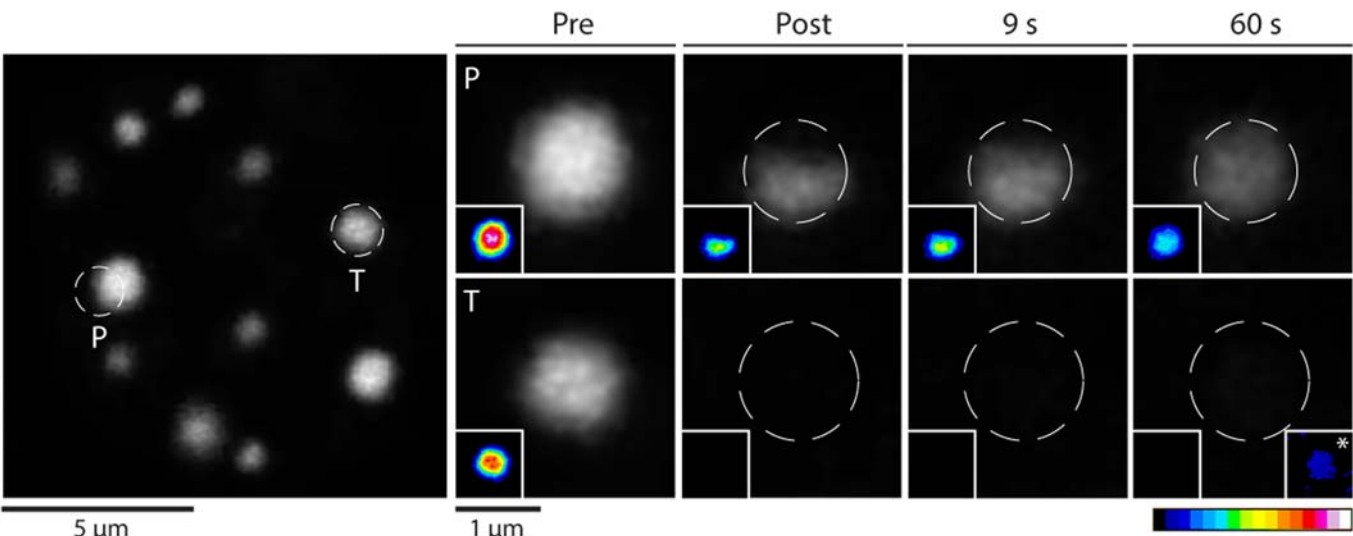

**Figure EV1.  Characterization of HRD dynamics within constitutive heterochromatin compartments.**

(A) Time-lapse series of total (T) and partial (P) HRD-mEmerald fluorescence recovery after photobleaching (Movie EV2) in mouse NMuMG immortalized breast epithelial cells (*n* = 45) (see Methods). Representative images are shown for pre-bleach, post-bleach (0.3 s), 9 s, and 60, and represent transiently transfected cells. Insets show fluorescence intensity using 16-color LUT. Asterisk denotes Hoechst channel to indicate presence of chromocenter.

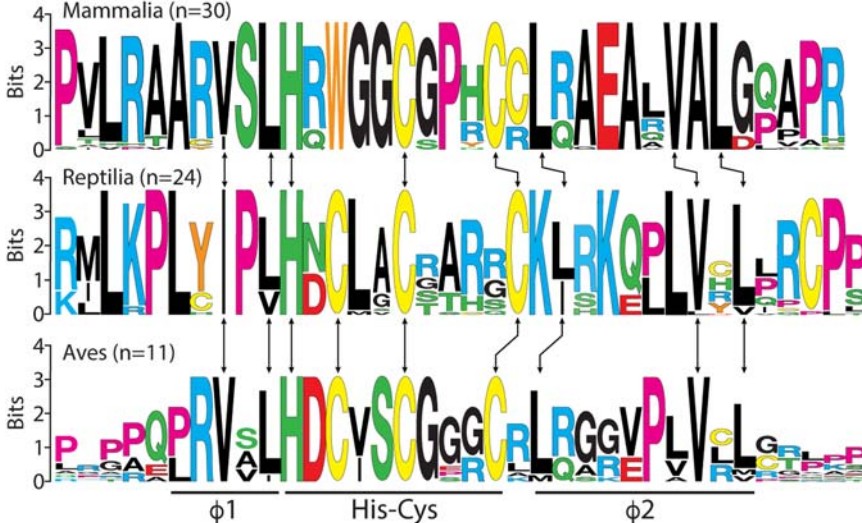

**Figure EV2. WebLogo depiction for the HRD CR1 from representative mammals, birds, and reptiles.**

The WebLogo (Crooks et al, 2004) is derived from the indicated number of sequences in each species. Key conserved features between classes are noted with arrows to account for differences in spacing. The frog version was omitted because the corresponding sequences show more variability in spacing between the histidine and cysteine residues. Detailed information regarding CR2 is provided in Fig. 5.

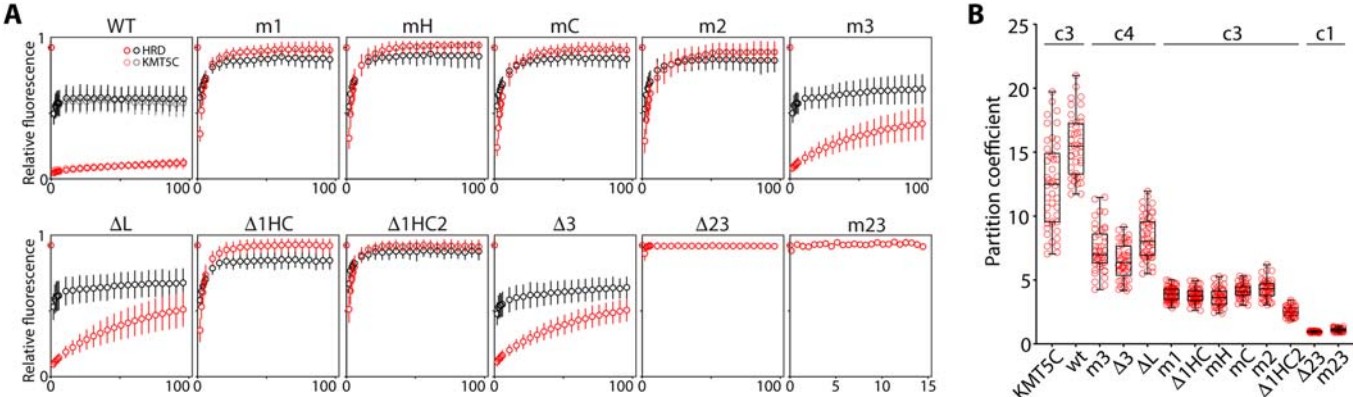

**Figure EV3. Complete FRAP and partitioning data for KMT5C, HRD, and corresponding derivatives.**

(A) FRAP curves (relative fluorescence vs. time) depict profiles for total (red) and partial (black) photobleaching of chromocenters for KMT5C and the HRD (superimposed), m1 (Φ1 mutant), mH (histidine mutant), mC (cysteine mutant), m2 (Φ2 mutant), m3 (Φ3 mutant), ΔL (linker deletion), Δ1HC (Φ1-histidine-cysteine deletion), Δ1HC2 (Φ1-histidine-cysteine-Φ2 deletion), Δ3 (Φ3 deletion), Δ23 (Φ2-Φ3 deletion), m23 (Φ2-Φ3 mutation). With the exception of m23, where recovery was monitored with higher sampling over a 15-second period, the remaining graphs show recovery from 0 to 100 s. (B) Partition coefficients (normalized chromocenter intensity divided by normalized nucleoplasmic intensity; see methods) are shown for the indicated proteins. Clusters (c1–c4) correspond to those in Fig. 3c. In FRAP plots, vertical lines correspond to the standard deviation of the mean. For box plots, vertical lines indicate the bounds of the box and whiskers (minimum to maximum) and the box corresponds to middle 50% of PC values (Q1–Q3) with the median indicated by a horizontal line. For FRAP and PC data, three separate experiments were conducted with a minimum of 15 cells each.

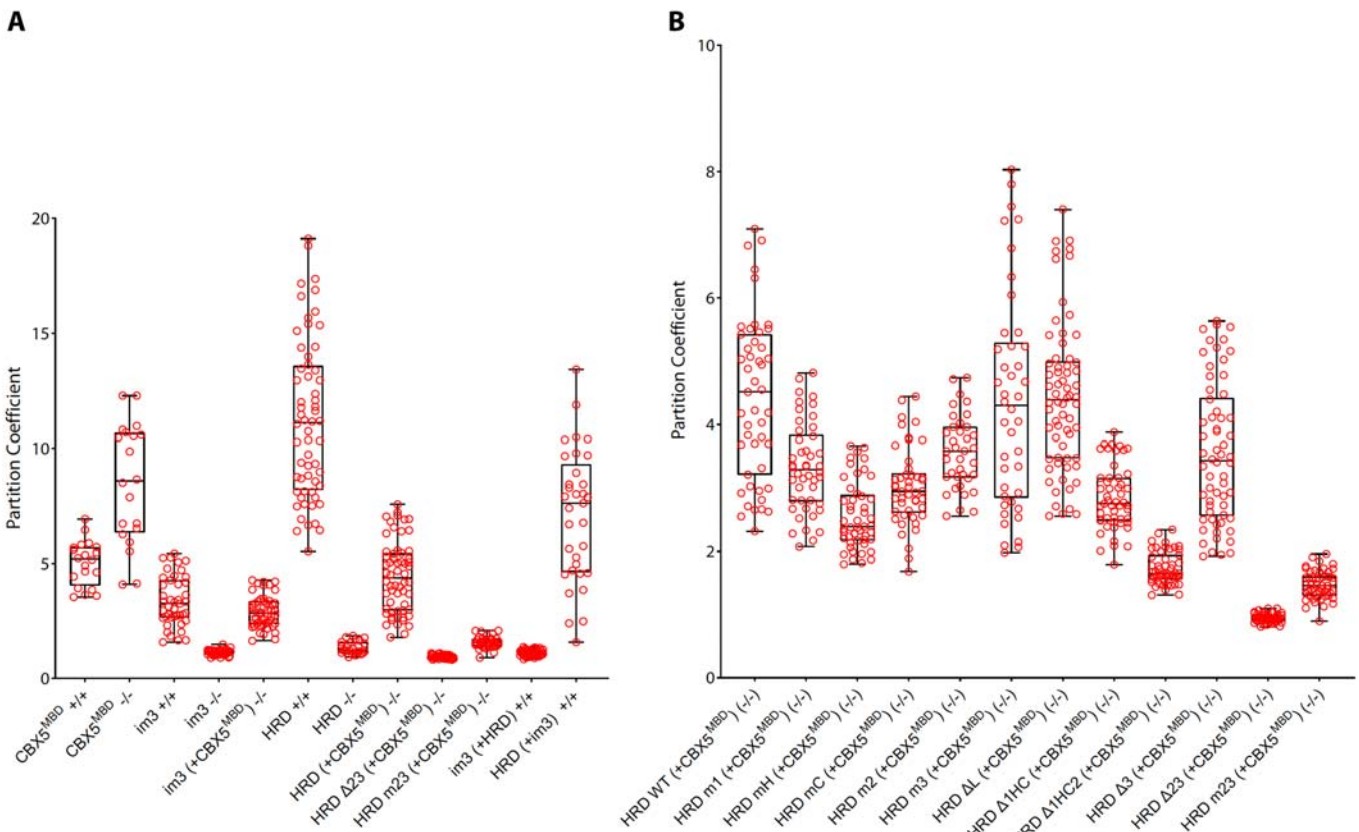

**Figure EV4. Partition coefficient data for *Suv39h1/2* (wild-type and null) and CBX5^MBD rescue experiments.**

(A) Partition coefficients (normalized chromocenter intensity divided by normalized nucleoplasmic intensity; see methods) are shown for the indicated proteins (single or co-transfection) and cell conditions (*Suv39h1/2* wild-type and null MEFs) from Fig. 5. (B) Partition coefficient data for CBX5^MBD rescue of all remaining mutant proteins from Fig. 3. For box plots, vertical lines indicate bounds of box and whiskers (minimum to maximum) and the box corresponds to middle 50% of PC values (Q1–Q3) with the median indicated by a horizontal line. For PC data, three separate experiments were conducted with a minimum of 15 cells each.

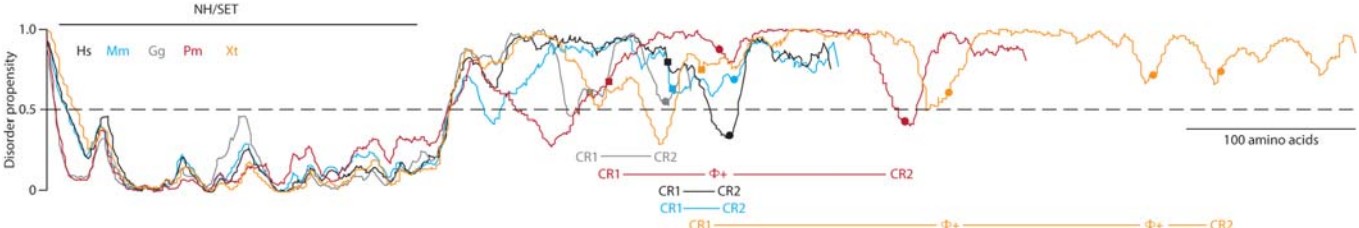

**Figure EV5. Disorder plot for representative KMT5C orthologs.**

Graph depicts Metapredict disorder propensity (0 being lowest and 1 being highest) for KMT5C orthologs from *Homo sapiens* (Hs), *Mus musculus* (Mm), *Gallus gallus* (Gg), *Podarcis muralis* (Pm), and *Xenopus tropicalis* (Xt) that have been anchored to the amino-terminal catalytic region (NH/SET). For each species, the location of the CR1 region is noted by a *square* and the CR2 region by a *circle*. For the longer Pm and Xt proteins, the presence of potential additional hydrophobic motifs (Φ+) in the region between the CR1 and CR2 motifs are noted.

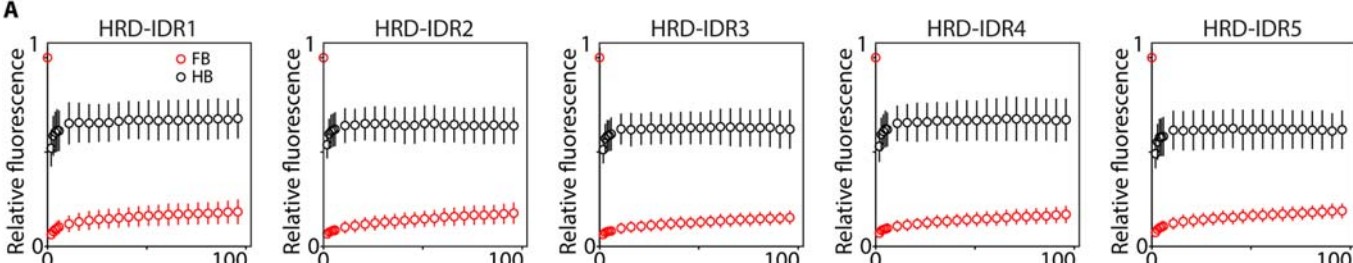

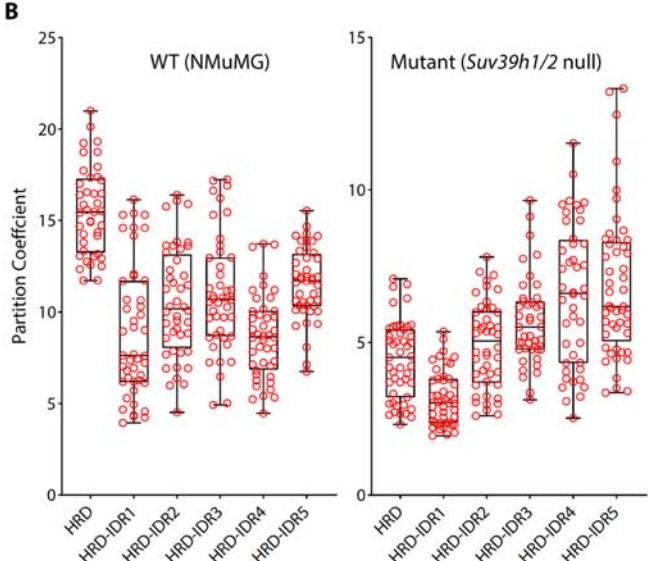

**Figure EV6. Complete FRAP and partitioning data for HRD chimeras.**

(A) FRAP curves (relative fluorescence vs. time) depict profiles for total (red) and partial (black) photobleaching for the HRD-IDR chimeras in NMuMG cells from 0 to 100 s. (B) Partition coefficients (normalized chromocenter intensity divided by normalized nucleoplasmic intensity; see methods) are shown for the indicated proteins in both wild-type and *Suv39h1/2* null conditions with co-expression of CBX5MBD. In FRAP plots, vertical lines correspond to the standard deviation of the mean. For box plots, vertical lines indicate the bounds of the box and whiskers (minimum to maximum) and the box corresponds to middle 50% of PC values (Q1–Q3) with the median indicated by a horizontal line. For FRAP and PC data, three separate experiments were conducted with a minimum of 15 cells each.

