## [Peer Review File · EMBO Reports]

KMT5C leverages disorder to optimize cooperation with HP1 for heterochromatin retention

Justin Knechtel, Hilmar Strickfaden, Kristal Missiaen, Joanne Hadfield, Michael Hendzel, and D Underhill

Corresponding author(s): D Underhill (underhil@ualberta.ca)

Review Timeline:

Submission Date:	3rd Feb 22
Editorial Decision:	11th Apr 22
Revision Received:	14th Jun 24
Editorial Decision:	19th Sep 24
Revision Received:	27th Sep 24
Accepted:	4th Nov 24

Editor: *Esther Schnapp*

Transaction Report:

Dear Dr. Underhill,

Thank you very much for your patience while your manuscript was peer-reviewed at EMBO reports, and I am sorry for the very unusual delay. It was difficult to find referees for your study, and I am glad that we have finally received two high quality reports that are pasted below.

As you will see, the referees acknowledge that the findings are potentially interesting. However, they both point out that the study is based on overexpression experiments, and that the physiological relevance of the findings remains unclear. Referee 1 further notes that the evidence for a LLPS-like behavior of KMT5C is insufficient.

Given these reports, it is clear that your study is a very borderline case. We can only proceed with its handling here if you can provide data that describe KMT5C behavior in a physiologically relevant context. In addition, LLPS of KMT5C can only be concluded if stronger data can be provided (along the lines suggested by referee 1). Otherwise, other models can be suggested, and should ideally be tested, but we can only proceed with your study here if the referees support the publication of your revised manuscript.

I am happy to offer a further discussion of your revisions, also by video chat, if you like. Just let me know in case you are interested and we can set an appointment.

Assuming that you are willing to address the referee concerns, I would like to invite you to revise your manuscript with the understanding that the referee concerns must be fully addressed and their suggestions taken on board. Please address all referee concerns in a complete point-by-point response. Acceptance of the manuscript will depend on a positive outcome of a second round of review. It is EMBO reports policy to allow a single round of major revision only and acceptance or rejection of the manuscript will therefore depend on the completeness of your responses included in the next, final version of the manuscript.

We realize that it is difficult to revise to a specific deadline. In the interest of protecting the conceptual advance provided by the work, we recommend a revision within 3 months (12th Jul 2022). Please discuss the revision progress ahead of this time with the editor if you require more time to complete the revisions.

- 1) A data availability section providing access to data deposited in public databases is missing. If you have not deposited any data, please add a sentence to the data availability section that explains that.
- 2) Your manuscript contains statistics and error bars based on $n=2$. Please use scatter blots in these cases. No statistics should be calculated if $n=2$.

3) We replaced Supplementary Information with Expanded View (EV) Figures and Tables that are collapsible/expandable online. A maximum of 5 EV Figures can be typeset. EV Figures should be cited as 'Figure EV1, Figure EV2' etc... in the text and their respective legends should be included in the main text after the legends of regular figures.

5) a complete author checklist, which you can download from our author guidelines <<https://www.embopress.org/page/journal/14693178/authorguide>>. Please insert information in the checklist that is also reflected in the manuscript. The completed author checklist will also be part of the RPF.

6) Please note that all corresponding authors are required to supply an ORCID ID for their name upon submission of a revised manuscript (<<https://orcid.org/>>). Please find instructions on how to link your ORCID ID to your account in our manuscript tracking system in our Author guidelines <<https://www.embopress.org/page/journal/14693178/authorguide#authorshipguidelines>>

8) We would also encourage you to include the source data for figure panels that show essential data. Numerical data should be provided as individual .xls or .csv files (including a tab describing the data). For blots or microscopy, uncropped images should be submitted (using a zip archive if multiple images need to be supplied for one panel). Additional information on source data and instruction on how to label the files are available at <<https://www.embopress.org/page/journal/14693178/authorguide#sourcedata>>.

- the name of the statistical test used to generate error bars and P values,
- the number (n) of independent experiments (please specify technical or biological replicates) underlying each data point,
- the nature of the bars and error bars (s.d., s.e.m.),
- If the data are obtained from n {less than or equal to} 2, use scatter blots showing the individual data points.

We would also welcome the submission of cover suggestions, or motifs to be used by our Graphics Illustrator in designing a

cover.

I look forward to seeing a revised form of your manuscript when it is ready.

Referee #1:

This manuscript provides a good demonstration that a HMT, KMT5c, targeting H3K20me concentrates in mouse chromocenters and diffuses within these chromocenters without diffusing significantly into the nucleoplasm over a time scale of many minutes. While they place this behavior in a qualitatively different category than other proteins such as HP1alpha and MeCP2, it looks like in fact this is more of a continuum of behaviors as shown by their data. They then go on to establish that this behavior is conferred by a protein region that is poorly conserved in sequence but distinct from intrinsic disordered regions (IDRs) that typically have been associated with LLPS.

I have two major issues with this manuscript, though, as written. They take a very heavy-handed approach, skewing all their results and discussion in terms of only a model of liquid-liquid phase separation and claiming that their results demonstrate a LLPS model. Instead, I would say they are describing several qualitative properties that are consistent but far from demonstrative of such a model. As Robert Tjian and colleagues reviewed recently, many papers particularly relating to papers involving live-cell imaging, have this problem.

I would say they have convincingly described the behavior of protein diffusion within the chromocenter without significant protein exchange (although see minor points below). On the other hand, their arguments about sphericity in their DNA damage experiments demonstrating LLPS are weak.

I don't believe the authors have come close to making such a conclusion about liquid like behavior, phase condensates, etc from their data, and therefore as written their experiments are not sufficient to back their conclusions. So they should rewrite their manuscript accordingly or provide additional experimental data in support of their conclusions.

For example, an alternative model that comes to mind immediately that would equally explain their major data of limited exchange of protein with the nucleoplasm but diffusion within the chromocenter would be a type of binding and release model in which the hopping between neighboring chromatin regions facilitates off rates of the protein. This type of model has been demonstrated by single molecule fluorescence approaches I believe for some proteins that in vivo and bulk biochemical methods had been described as rapidly exchanging by FRAP and other measurements. More specifically, DNA binding proteins that by FRAP appear as rapidly exchanging, coming on and off the DNA in seconds, have been shown to be stably bound at low DNA concentration. However, when DNA molecules are added to the solution, transfer of protein off of the original DNA strand is facilitated through transfer to a new DNA molecule. Under these conditions off rates are rapid and on a time scale of seconds. (for example, see ERBAS, A., MARKO, J.F. Current Opinion in Chemical Biology, 2019, 53, pp. 118-124)

Thus an alternative model that has nothing to do with condensates or LLPS would be one in which the protein transfers sequentially within the chromocenter through such a binding and transfer and/or hopping type action. That would seem to make more sense with regard to their demonstration of a bipartite protein domain that confers this behavior and segregation and mixing within the chromocenter that is unrelated to IDR content. For instance, if they knocked down or out KMT5C and then expressed the protein at a very low level (below the critical concentration for phase separation) would they still see the same behavior of retention within the chromocenter at a protein concentration too low to form a condensate?

If the authors want to conclude an explanation involving their LLPS they should test and rule out alternative models and/or provide stronger support through additional experimentation. One stronger demonstration of LLPS behavior would be if they could demonstrate classic behavior specific to first order phase transitions. For example a test that the Brandwynne lab has done is to examine behavior as a function of increasing protein expression. If this is a true condensate/LLPS behavior they should see at low expression level a nucleoplasmic diffuse staining until they reach a critical concentration. At that point the intensity within the condensate and total condensate should increase while the nucleoplasmic concentration stays constant. This would require them to deplete endogenous KMT5C and then express varying and increasing amounts of the protein.

Alternatively, the authors could add SPT to support true liquid-like behavior and rule out the type of facilitated transfer model proposed above.

A second significant problem that I have is that it is entirely based on plasmid transgene overexpression. In my own laboratory, we have done similar transient expression of over a dozen nuclear proteins with significant IDR content. Nearly all of them show very "liquid-like" condensates in live cells in the vast majority of transfected cells. However, this behavior is clearly correlated with expression level with a small fraction of low expressing cells showing qualitatively different protein distributions that actually correspond to what we see after antibody staining of the endogenous protein or knock in of the fluorescent protein tag into the endogenous gene locus. There doesn't seem to have been any attention to this issue, and it may be particularly relevant to statements about the appearance of sphericity, etc, particularly as the authors are not doing colocalization relative to DNA labeling and other chromocenter markers and/or showing careful measurement of chromocenter features in comparison to wild type cells.

Thus, I really believe that every significant study now done in mammalian cells needs to use methods for some subset of experiments to express proteins at more physiological levels. There have been a wave of papers published in the last few years about LLPS that I believe are physiologically irrelevant because the condensates described are artifacts of overexpression. We now study several proteins in my lab that form round, LLPS when overexpressed but form completely different structures at physiological levels of expression. Traditional methods for this are to use knock-in tagging of endogenous genes or, previously, using BAC transgenes that express within several fold of endogenous genes (see work from Neugebauer lab for example). Previous to this, in several areas of cell biology, there have been clear examples in which well-studied phenomenon showed completely different behavior once protein expression was at physiological levels. An example is work from the Drubin lab in which they resolved a long-standing discrepancy between mammalian and yeast cells in the area of protein targeting simply by using CRISPR tagging of endogenous genes instead of transgenes to express the same proteins. Once they used endogenous protein tagging, results from yeast and mammalian systems matched exactly, whereas the field had debated about apparent differences between these two systems for years before this.

Minor Points:

1. They should discuss more in the main text about how they do measurements to support statements. As a simple example, in Movie 1 the intensities of the different time points after bleaching do not change. They are arguing that protein flux into the chromocenter is not happening- only molecules diffusing from fluorescent to bleached regions within the chromocenter. That would imply total fluorescence remains the same so intensities should drop over time as the protein spreads out over the chromocenter. But the images do not show this- I measured in ImageJ. So are they normalizing time points or does their data not support their model.
2. Along the same line, are they accurately measuring intensities. Is zero in their image really the intensity corresponding to zero light- or something else. Confocal raster scan microscopes notoriously arbitrarily set black to whatever level the user chooses leading to nonlinear measurements. Here they used a spinning disk microscope but what did they do to establish the proper scaling of intensity?

In the main text they did not deal with this adequately in Results to explain to the reader trying to follow the story.

3. Continuing along the same line, in the first main figures it appears that there is no fluorescent nucleoplasmic exchange yet in one of the supplementary figures in which they bleach half the nucleus there is clearly some nucleoplasmic exchange. So this is a quantitative issue and that explains the cartoons showing exchange rate comparisons between the different studied proteins.

I didn't take the time to try to dig this out of the Methods. Again, without going into details of HOW they did this, in the main Results section they should at least describe the measurements and modeling and major results of this modeling. I got the feeling that the differences among the KMT5C, MeCPT, and HP1alpha were more quantitative variations along a range rather than qualitatively different behaviors.

Referee #2:

Summary

In this paper, the authors characterize the interplay between sequence-features and emergent dynamics of KMT5C, a lysine methyltransferase protein, in the context of heterochromatin (HC) (or chromocenter) condensates compartmentalized through phase separation. First, the authors develop a Fluorescence Recovery after Photobleaching (FRAP) assay in mouse-derived cell lines to characterize dynamics of key HC proteins - HP1 (CBX5), MECP2, and KMT5C. By comparing assays on partial (internal rearrangement) and complete (exchange with nucleoplasm) bleach experiments - they find that unlike classic HC markers (CBX5, MECP2) which are simultaneously highly mobile and exchange with surroundings, KMT5C exhibits rapid internal dynamics (within a chromocenter or condensate) but extremely limited exchange with surroundings. By introducing DNA damage at individual chromocenters, the authors characterize how KMT5C in chromocenters split and round-up into fission-induced compartments. Finally, through sequence-analyses and imaging experiments, they identify conserved features, that are distributed across a specific domain as short motifs, that encode for the observed dynamics and lack of exchange. Interestingly, they note that different linkers between these motifs may tune dynamical properties but not localization or exchange.

Together, this paper reports interesting and new data, is well-written, and clearly delineates a sequence-phase-behavior relationship for KMT5C in the context of HC condensates. Overall, this reviewer is excited by the key message outlined in this study but feels there are few issues that remain unanswered or are unclear as currently written. If addressed, through experiments, analyses, and/or writing, this paper would be much improved for publication.

Major Concerns:

Role of expression-levels in chromocenter formation: As the authors clearly recognize and state, phase separation is a highly concentration dependent process. The assays to probe chromocenters in this paper are based on transient transfection and not at endogenous levels of protein. Do the authors believe that this does not directly affect much of observed data? For example: Are the number of chromocenters similar in cells with transfected assays as measured by other assays that don't change concentration? (antibody assays for key proteins or dense DNA or pertinent histone marks or live-cell imaging come to mind as potential alternatives)

Relatedly, the methods and text are not very clear about the specific molecules (plasmids), tags (fluorescence), and levels reported - some of which is in the supplementary. For example, nowhere is it stated in the main text that the imaging is done on cells expressing transiently transfected proteins nor the tag. Clarifying the text in this regard would be helpful for the readers. Have the authors performed any colocalization assays? Is the imaging assay for each protein measuring the same, overlapping set of HC condensates? Do some HC condensates have higher amounts of KMT5C? What is the relative partitioning of multiple species in the same cell? Even reporting statistics on volume, number, fraction etc. across modalities can lend confidence that the authors are studying the same condensate across experiments.

DNA-damage assays: The micro-irradiation assays, while showing a clearly reported phenotype for KMT5C, raises a number of questions that make interpretation of the data hard (especially with only experimental data reported per condition) and hard to reconcile the claims in this section:

If there is chromatin opening up (and formation of a DNA-damage response condensate, as the author suggests)  does their data then suggest a picture in which a DNA damage response condensate forms but selectively partitions HP1, MECP2 but not KMT5C? If so, this is a strong claim that needs more validation.

How can you be confident that the same DNA-damage response is being elicited across conditions? Staining for a prototypical marker would go a long way in supporting this hypothesis. Otherwise, an argument may be that irradiation does not induce fission in the conditions reported for MECP2 or HP1.

Ideally, additional replicates of the experiment (reported through statistical analyses of (1) number of fission events, (2) sphericity and (3) DNA staining?)

CRD analyses: These experiments are really interesting but a few gaps that are puzzling are listed below.

Can the authors report their phenotype that CRD is sufficient for chromocenter localization and dynamics? For e.g. is a version of the protein without CRD not able to localize? Including the RBD+CRD mutant data in the supplementary would be helpful instead of listing as (data not shown).

For the mutants explored- is there a reason that instead of standard Alanine scanning (where every residue is mutated to A) - some are mutated to Leucines or Asparagines? How is their data dependent on this choice of assay? The inconsistency in mutants (barring technical difficulties which the authors are welcome to have a brief discussion and explanation of in the manuscript) complicates interpretation of the data.

Finally, as a note, it would add further strength to explore the partitioning (readily accessible from your existing data) analyses of the different mutants as well as linker IDRs. These can help compare all the different proteins/domains on the same set of relevant axes, which are (partitioning, internal mobility, and exchange).

In fact, a graph of different domains and proteins with partitioning vs internal mobility (from an "apparent" diffusion) or exchange-rate (rate-of-recovery from full-bleach) would be a helpful and vivid summary of the many different experiments and I

recommend the authors to consider this.

Minor Concerns:

The acronym CRD is used multiple times before being defined - please fix this.

Figure 6, particularly 6B, is challenging to interpret. I don't have suggestions to improve but would appreciate a clearer presentation and leave it to the authors discretion.

In the FRAP analyses, is the background always normalized to unbleached regions in the same cell? I am particularly interested in the inverse FRAP experiments that report an efflux rate of 0.7% → since the normalization, if done, may come from a different cell, how can the authors be confident this represents exchange and not fluorophore loss? In any case, a brief sentence or two would be useful in the main text.

Can you report the apparent diffusivity of KMT5C in the nucleus? This would be a useful contrast number to consider in the context of measuring relative difference in/out of condensate.

Author responses

We thank the reviewers for their thoughtful evaluation of the manuscript and provide a detailed response below. Although this manuscript was prepared under the assumption that it would be treated as a new submission, it was re-categorized with a request to respond to the previous reviews. The direction this revised version took was in no small part due to the valuable comments of the reviewers, for which we express our gratitude. While the revised version begins with the same observation relating to KMT5C dynamics within chromocenters, it now addresses the mechanistic basis with a direct evaluation of the role intrinsic disorder plays in the behavior. This includes leveraging additional sequence annotation to enable comprehensive mutagenesis and deletion analyses, as well as providing further insight to the role of HP1 in conferring the retentive behavior of KMT5C. A major issue raised by both reviewers in the prior submission pertained to the focus on liquid-liquid phase separation (LLPS). This has effectively been removed from the resubmitted manuscript and is addressed in the discussion in the context of other more plausible explanations. This reflects the challenges with definitively establishing LLPS in live cells, as well as new data that are consistent with binding/hopping mechanism mentioned by reviewer 1. Central to this idea is that we now relate KMT5C partitioning and dynamics to expression levels, and demonstrate chromocenter saturability. Because of these changes, we have removed data that, while interesting to KMT5C biology, is no longer pertinent to the narrative. These instances are specifically noted in the responses below and primarily relate to laser micro-irradiation and morphological assays. We would also like to emphasize that despite the time that has transpired, the observation continues to be highly novel.

Reviewer comments shown in **bold**.

Reviewer #1

"This manuscript provides a good demonstration that a HMT, KMT5c, targeting H4K20me concentrates in mouse chromocenters and diffuses within these chromocenters without diffusing significantly into the nucleoplasm over a time scale of many minutes. While they place this behavior in a qualitatively different category than other proteins such as HP1alpha and MeCP2, it looks like in fact this is more of a continuum of behaviors as shown by their data. They then go on to establish that this behavior is conferred by a protein region that is poorly conserved in sequence but distinct from intrinsic disordered regions (IDRs) that typically have been associated with LLPS.

I have two major issues with this manuscript, though, as written. They take a very heavy-handed approach, skewing all their results and discussion in terms of only a model of liquid-liquid phase separation and claiming that their results demonstrate a LLPS model. Instead, I would say they are describing several qualitative properties that are consistent but far from demonstrative of such a model. As Robert Tjian and colleagues reviewed recently, many papers particularly relating to papers involving live-cell imaging, have this problem."

1. **I would say they have convincingly described the behavior of protein diffusion within the chromocenter without significant protein exchange (although see minor points below).**
2. **On the other hand, their arguments about sphericity in their DNA damage experiments demonstrating LLPS are weak.**
 - a. We agree. While they may be relevant to unrelated aspects of KMT5C in DNA damage, we recognize that their potential relationship to LLPS was indirect and tenuous. Accordingly, these experiments are no longer included in our revised manuscript.
3. **I don't believe the authors have come close to making such a conclusion about liquid like behavior, phase condensates, etc. from their data, and therefore as written their experiments are not sufficient to back their conclusions. They should rewrite their manuscript accordingly or provide additional experimental data in support of their conclusions.**
 - a. We agree and have rewritten the manuscript in line with the suggestions. To that end, we have acquired additional data that suggests an alternative model for KMT5C dynamics that better fits with a binding and release mechanism as suggested by the reviewer in the next comment. In addition, recent studies of HP1 (PubMed IDs 36526633 and 32101700) and MECP2 (PubMed ID 38719804) also describe models based on chromatin binding sites (either histone modifications or DNA methylation) as drivers of localization independent of LLPS.

4. For example, an alternative model that comes to mind immediately that would equally explain their major data of limited exchange of protein with the nucleoplasm but diffusion within the chromocenter would be a type of binding and release model in which the hopping between neighboring chromatin regions facilitates off rates of the protein. This type of model has been demonstrated by single molecule fluorescence approaches I believe for some proteins that in vivo and bulk biochemical methods had been described as rapidly exchanging by FRAP and other measurements. More specifically, DNA binding proteins that by FRAP appear as rapidly exchanging, coming on and off the DNA in seconds, have been shown to be stably bound at low DNA concentration. However, when DNA molecules are added to the solution, transfer of protein off the original DNA strand is facilitated through transfer to a new DNA molecule. Under these conditions off rates are rapid and on a time scale of seconds (for example, see ERBAS, A., MARKO, J.F. Current Opinion in Chemical Biology, 2019, 53, pp. 118-124).
 - a. We have updated our model in the revised manuscript and, as noted above, our current data are in line with the bind and release model suggested by the reviewer.
5. Thus, an alternative model that has nothing to do with condensates or LLPS would be one in which the protein transfers sequentially within the chromocenter through such a binding and transfer and/or hopping type action. That would seem to make more sense regarding their demonstration of a bipartite protein domain that confers this behavior and segregation and mixing within the chromocenter that is unrelated to IDR content. For instance, if they knocked down or out KMT5C and then expressed the protein at a very low level (below the critical concentration for phase separation) would they still see the same behavior of retention within the chromocenter at a protein concentration too low to form a condensate?
 - a. Adding to the previous response, we now assess concentration-dependence for both FRAP and partitioning (Figure 4B) and document a decrease in retention due to saturability of binding sites. This also shows 'sink-like' accumulation and no evidence for a critical concentration, further arguing against LLPS.
6. If the authors want to conclude an explanation involving their LLPS they should test and rule out alternative models and/or provide stronger support through additional experimentation. One stronger demonstration of LLPS behavior would be if they could demonstrate classic behavior specific to first order phase transitions. For example, a test that the Brangwynne lab has done is to examine behavior as a function of increasing protein expression. If this is a true condensate/LLPS behavior, they should see at low expression level a nucleoplasmic diffuse staining until they reach a critical concentration. At that point the intensity within the condensate and total condensate should increase while the nucleoplasmic concentration stays constant. This would require them to deplete endogenous KMT5C and then express varying and increasing amounts of the protein.
 - a. As above, our data show behaviors that are not consistent with LLPS and the manuscript has been revised to indicate this.
7. Alternatively, the authors could add SPT to support true liquid-like behavior and rule out the type of facilitated transfer model proposed above.
 - a. We agree that SPT on KMT5C will be valuable and have initiated those experiments. As mentioned, our current narrative does carefully document concentration effects and the limits created by exogenous expression.
8. A second significant problem that I have is that it is entirely based on plasmid transgene overexpression. In my own laboratory, we have done similar transient expression of over a dozen nuclear proteins with significant IDR content. Nearly all of them show very "liquid-like" condensates in live cells in the vast majority of transfected cells. However, this behavior is clearly correlated with expression level with a small fraction of low expressing cells showing qualitatively different protein distributions that actually correspond to what we see after antibody staining of the endogenous protein or knock in of the fluorescent protein tag into the endogenous gene locus. There doesn't seem to have been any attention to this issue, and it may be particularly relevant to statements about the appearance of sphericity, etc., particularly as the authors are not doing colocalization relative to DNA labeling and other chromocenter markers and/or showing careful measurement of chromocenter features in comparison to wild type cells.

- a. We acknowledge that using solely plasmid transgene overexpression is a limitation and have addressed this point in the paper. We also more carefully relate the retentive behavior to expression levels. In addition, we highlight that one of the original papers on KMT5C mobility (Hahn, 2013; PubMed ID 23599346) did use a GFP knock-in allele and noted the same lack of exchange (but did not assess internal mobility). The retentive behavior has therefore been established at endogenous KMT5C levels. Moreover, the distribution of KMT5C is concordant with numerous studies showing marked enrichment of its enzymatic product (H4K20me3) in chromocenters.
9. **Thus, I really believe that every significant study now done in mammalian cells needs to use methods for some subset of experiments to express proteins at more physiological levels. There have been a wave of papers published in the last few years about LLPS that I believe are physiologically irrelevant because the condensates described are artifacts of overexpression. We now study several proteins in my lab that form round, LLPS when overexpressed but form completely different structures at physiological levels of expression. Traditional methods for this are to use knock-in tagging of endogenous genes or, previously, using BAC transgenes that express within several fold of endogenous genes (see work from Neugebauer lab for example). Previous to this, in several areas of cell biology, there have been clear examples in which well-studied phenomenon showed completely different behavior once protein expression was at physiological levels. An example is work from the Drubin lab in which they resolved a long-standing discrepancy between mammalian and yeast cells in the area of protein targeting simply by using CRISPR tagging of endogenous genes instead of transgenes to express the same proteins. Once they used endogenous protein tagging, results from yeast and mammalian systems matched exactly, whereas the field had debated about apparent differences between these two systems for years before this.**
- a. As noted above, retentive behavior has been shown at endogenous protein levels and we have shown that this persists over a wider concentration range until it reaches saturation. As part of this analysis, we show that the unique retentive behaviour of KMT5C is most apparent at low expression levels. These points are discussed in the resubmission.

Minor points:

1. **They should discuss more in the main text about how they do measurements to support statements. As a simple example, in Movie 1 the intensities of the different time points after bleaching do not change. They are arguing that protein flux into the chromocenter is not happening- only molecules diffusing from fluorescent to bleached regions within the chromocenter. That would imply total fluorescence remains the same so intensities should drop over time as the protein spreads out over the chromocenter. But the images do not show this- I measured in ImageJ. So, are they normalizing time points or does their data not support their model?**

- a. With analyses of multiple chromocenters/cells, we do consistently see fluorescence loss in the unbleached portion of the chromocenter and a corresponding recovery in the bleached area (see adjacent graph). The kymograph in figure 1A also shows that KMT5C redistributes within the chromocenter until reaching a plateau as shown with the intensity plot.

2. **Along the same line, are they accurately measuring intensities? Is zero in their image really the intensity corresponding to zero light- or something else. Confocal raster scan microscopes notoriously arbitrarily set black to whatever level the user chooses leading to nonlinear measurements. Here they used a spinning disk microscope but what did they do to establish the proper scaling of intensity? In the main text they did not deal with this adequately in Results to explain to the reader trying to follow the story.**

- a. Images are background corrected and normalized to account for photobleaching from scanning. This information has been added to the methods section of the main text.
3. **Continuing along the same line, in the first main figures it appears that there is no fluorescent nucleoplasmic exchange yet in one of the supplementary figures in which they bleach half the nucleus there is clearly some nucleoplasmic exchange. So, this is a quantitative issue and that explains the cartoons showing exchange rate comparisons between the different studied proteins.**
 - a. This is addressed by our experimental results in figure 4B. These results indicate that the retention of KMT5C is highly dependent on concentration, which we have now controlled for and applied to our interpretation and model.
10. **I didn't take the time to try to dig this out of the Methods. Again, without going into details of HOW they did this, in the main Results section they should at least describe the measurements and modeling and major results of this modeling. I got the feeling that the differences among the KMT5C, MECP2, and HP1alpha were more quantitative variations along a range rather than qualitatively different behaviors.**
 - a. We agree and have omitted the comparisons in the resubmission. As addressed in major point 3, the idea of quantitative variations also fits with recent work on HP1 and MECP2 that suggest localization is driven by binding site availability and not LLPS.

Reviewer #2

In this paper, the authors characterize the interplay between sequence-features and emergent dynamics of KMT5C, a lysine methyltransferase protein, in the context of heterochromatin (HC) (or chromocenter) condensates compartmentalized through phase separation. First, the authors develop a Fluorescence Recovery after Photobleaching (FRAP) assay in mouse-derived cell lines to characterize dynamics of key HC proteins - HP1 (CBX5), MECP2, and KMT5C. By comparing assays on partial (internal rearrangement) and complete (exchange with nucleoplasm) bleach experiments - they find that unlike classic HC markers (CBX5, MECP2) which are simultaneously highly mobile and exchange with surroundings, KMT5C exhibits rapid internal dynamics (within a chromocenter or condensate) but extremely limited exchange with surroundings. By introducing DNA damage at individual chromocenters, the authors characterize how KMT5C in chromocenters split and round-up into fission-induced compartments. Finally, through sequence-analyses and imaging experiments, they identify conserved features, that are distributed across a specific domain as short motifs, that encode for the observed dynamics and lack of exchange. Interestingly, they note that different linkers between these motifs may tune dynamical properties but not localization or exchange.

Together, this paper reports interesting and new data, is well-written, and clearly delineates a sequence-phase-behavior relationship for KMT5C in the context of HC condensates. Overall, this reviewer is excited by the key message outlined in this study but feels there are few issues that remain unanswered or are unclear as currently written. If addressed, through experiments, analyses, and/or writing, this paper would be much improved for publication.

1. **Role of expression-levels in chromocenter formation: As the authors clearly recognize and state, phase separation is a highly concentration dependent process. The assays to probe chromocenters in this paper are based on transient transfection and not at endogenous levels of protein. Do the authors believe that this does not directly affect much of observed data? For example:**
2. **Are the number of chromocenters similar in cells with transfected assays as measured by other assays that don't change concentration? (Antibody assays for key proteins or dense DNA or pertinent histone marks or live-cell imaging come to mind as potential alternatives).**
 - a. As noted in the response to reviewer 1, we acknowledge that using plasmid transgene overexpression is a limitation and have carried out additional studies that address retentive behavior as a function of expression level where we document saturability. Likewise, we do emphasize that earlier work with an endogenous GFP knock-in allele at the *Kmt5c* allele was characterized by lack of exchange, although internal mobility was not assessed at that time.
 - b. We have not currently assessed number of chromocenters and it is possible that KMT5C levels could alter chromatin structure. In that regard, the ability for KMT5C to alter chromocenter number and morphology has been previously shown (Hahn, 2013; PubMed ID 23599346).

3. **Relatedly, the methods and text are not very clear about the specific molecules (plasmids), tags (fluorescence), and levels reported - some of which is in the supplementary. For example, nowhere is it stated in the main text that the imaging is done on cells expressing transiently transfected proteins nor the tag. Clarifying the text in this regard would be helpful for the readers.**
 - a. We agree that this is important for clarity and have added this information to the methods section of main text and relevant figure legends.
4. **Have the authors performed any colocalization assays? Is the imaging assay for each protein measuring the same, overlapping set of HC condensates? Do some HC condensates have higher amounts of KMT5C? What is the relative partitioning of multiple species in the same cell? Even reporting statistics on volume, number, fraction etc. across modalities can lend confidence that the authors are studying the same condensate across experiments.**
 - a. The additional experimental data included in our revised manuscript includes detailed analysis of relative partitioning of various proteins across multiple cell lines and as a function of expression level. In this context, we also establish the dependence on HP1 and H3K9me3 binding in colocalization and rescue experiments.
5. **DNA-damage assays: The micro-irradiation assays, while showing a clearly reported phenotype for KMT5C, raises a number of questions that make interpretation of the data hard (especially with only experimental data reported per condition) and hard to reconcile the claims in this section:**
 - a. The DNA damage assays are no longer included in our revised manuscript.
6. **If there is chromatin opening up (and formation of a DNA-damage response condensate, as the author suggests)  does their data then suggest a picture in which a DNA damage response condensate forms but selectively partitions HP1, MECP2 but not KMT5C? If so, this is a strong claim that needs more validation. How can you be confident that the same DNA-damage response is being elicited across conditions? Staining for a prototypical marker would go a long way in supporting this hypothesis. Otherwise, an argument may be that irradiation does not induce fission in the conditions reported for MECP2 or HP1.**
 - a. The DNA damage assays are no longer included in our revised manuscript.
7. **Ideally, additional replicates of the experiment (reported through statistical analyses of (1) number of fission events, (2) sphericity and (3) DNA staining?)**
 - a. All experimental data was replicated at least 3 times with at least 45 individual cells. This information is now included in the main text where relevant. Fission events (relating to laser micro-irradiation) and sphericity are no longer part of the revised manuscript.
8. **CRD analyses: These experiments are really interesting but a few gaps that are puzzling are listed below. Can the authors report their phenotype that CRD is sufficient for chromocenter localization and dynamics? For e.g., is a version of the protein without CRD not able to localize? Including the RBD+CRD mutant data in the supplementary would be helpful instead of listing as (data not shown).**
 - a. The revised manuscript has significantly more sequence and mutational analysis, which we believe addresses these concerns. We also renamed the domain to heterochromatin retention domain (HRD). All motifs are subject to both point and deletion mutagenesis, which has provided much more clarity on mechanism of action and that the HRD can recapitulate the behavior of the full-length KMT5C protein.
9. **For the mutants explored- is there a reason that instead of standard Alanine scanning (where every residue is mutated to A) - some are mutated to Leucines or Asparagines? How is their data dependent on this choice of assay? The inconsistency in mutants (barring technical difficulties which the authors are welcome to have a brief discussion and explanation of in the manuscript) complicates interpretation of the data.**
 - a. For each mutation, we selected the most conservative substitution possible to avoid substantive changes in physicochemical properties and confounding phenotypes. For instance, we did find that the retentive behavior of the HRD was very sensitive to the presence of negative charge in the linker. This explanation and rationale are included in the methods section of the main text. We provide a more detailed description and discussion of HRD composition and its role in retention.

10. Finally, as a note, it would add further strength to explore the partitioning (readily accessible from your existing data) analyses of the different mutants as well as linker IDRs. These can help compare all the different proteins/domains on the same set of relevant axes, which are (partitioning, internal mobility, and exchange). In fact, a graph of different domains and proteins with partitioning vs internal mobility (from an "apparent" diffusion) or exchange-rate (rate-of-recovery from full-bleach) would be a helpful and vivid summary of the many different experiments and I recommend the authors to consider this.

- a. We agree and have included additional partitioning analyses in our revised manuscript, which includes assessing concentration-dependence. A significant change in the revised version is more detailed analyses of the linker regions, which has provided considerable insight to mechanism of action. We include visual cues in the figure and have substantively revised the data presentation to improve clarity, as well as provide an overarching model.

Minor concerns

1. The acronym CRD is used multiple times before being defined - please fix this.

- a. This is addressed in our revised manuscript. As noted, we have revised the nomenclature to refer to this as the HRD.

2. Figure 6, particularly 6B, is challenging to interpret. I don't have suggestions to improve but would appreciate a clearer presentation and leave it to the authors discretion.

- a. This has been largely reconfigured in the revised manuscript and we hope it now provides a clearer presentation.

3. In the FRAP analyses, is the background always normalized to unbleached regions in the same cell? I am particularly interested in the inverse FRAP experiments that report an efflux rate of 0.7% → since the normalization, if done, may come from a different cell, how can the authors be confident this represents exchange and not fluorophore loss? In any case, a brief sentence or two would be useful in the main text. Can you report the apparent diffusivity of KMT5C in the nucleus? This would be a useful contrast number to consider in the context of measuring relative difference in/out of condensate.

- a. Yes, background is typically normalized to the same cell. Further details have been added to the live cell imaging methods description.
- b. For the iFRAP experiment, we considered the entire cell as the bleached region and corrected using the background outside of the cell in calculations. For the issue of photobleaching as a result of repetitive imaging noted by the reviewer, fluorescence of the ROI was normalized to the fluorescence of the whole cell. In addition, the saturation analyses included in the revised paper supports the idea of 'sink-like' accumulation of the HRD over an extended range of expression levels. This information is now provided in the text.
- c. We can provide an estimate of the difference between chromocenters and the nucleoplasm below. Using the exchange rate from iFRAP as a proxy for $t_{1/2}$, it would require 71 minutes (4260s) for half recovery to take place. From nucleoplasmic FRAP, we estimate a $t_{1/2}$ of 36.19s, which corresponds to a 118-fold difference in values. Previous analyses by Müller-Ott, 2014; PubMed ID 25134515) identified similar differences in recovery for KMT5C when comparing heterochromatin to euchromatin. For reference, the recovery curve for the HRD in chromocenters vs. nucleoplasm is below.

Dear Alan,

Thank you for the submission of your revised manuscript. We have now received the enclosed reports from the referees that were asked to assess it. Referee 1 was unfortunately not available to re-review your ms and I therefore added a new referee 3. Both referees still have a few more minor suggestions that I would like you to incorporate before we can proceed with the official acceptance of your manuscript.

A few editorial requests will also need to be addressed:

- Please submit the final ms also as a word file without figures.
- Please add up to 5 keywords to the ms.
- The 'Containing data deposition statement' needs to be renamed to "Data Availability Section" (DAS) and needs to have some text. If no data were generated in this study and submitted to public databases please mention this fact in the DAS.
- The conflict of interest heading needs to be renamed to "Disclosure Statement and Competing Interests"
- Please place the author affiliations below the author list on the ms title page.
- The author credits need to be removed from the ms files. All credits are entered during online ms submission.
- Please correct the REFERENCE FORMAT the the EMBO reports (Harvard) style. It needs to be alphabetical, not numerical; et al needs to be used after 10 author names; DOIs should only be used for preprints and datasets that have not been published yet.
- DATA NOT SHOWN on page 11 needs to be removed as per journal policy.
- Please send us a completed author checklist with your final ms, which you can download from our author guidelines <<https://www.embopress.org/page/journal/14693178/authorguide>>. The completed author checklist will also be part of the transparent peer-review file.
- Please upload all main and EV figures as separate production high quality Figure files.
- Table S3 is called out but it is missing; Movies 15-18 are not called out, please correct.
- Table S1 and Table S2 both seem to be Datasets. It would be better to upload these separately as Dataset EV1 and Dataset EV2; each legend should be provided in its corresponding Excel file as a separate tab/sheet; ms callouts need to be adapted accordingly.
- The supplementary Figures S1-S6 can be placed in a PDF Appendix file with a ToC on the title page; the nomenclature would then need to be Appendix Figure S1-S6; these 6 figures could also be EV figures - the legends could stay in the ms, while each figure should be upldd as a separate Figure file: Figure EV1-EV6. You can find more info on our file types in our guide to authors online: <https://www.embopress.org/page/journal/14693178/authorguide#expandedview>
- The movie nomenclature should be corrected: source file names, titles in eJP and ms callouts to Movie EV1-EV19; each movie needs a legend provided as a readme.txt file and then each movie should be zipped up with its legend and upldd as one zip folder per movie.
- The Methods section should include a Reagents and Tools Table (listing key reagents, experimental models, software and relevant equipment and including their sources and relevant identifiers) followed by a Methods and Protocols section in which we encourage the authors to describe their methods using a step-by-step protocol format with bullet points, to facilitate the adoption of the methodologies across labs. More information on how to adhere to this format as well as downloadable templates (.docx) for the Reagents and Tools Table can be found in our author guidelines: <<https://www.embopress.org/page/journal/14693178/authorguide#manuscriptpreparation>>
- Materials and methods should be "Methods"
- The manuscript sections should be in the following order: Title page - Abstract & Keywords - Introduction - Results - Discussion - Methods - Data Availability - Acknowledgments - Disclosure Statement & Competing Interests - References - Figure Legends - (Main Tables with legends if applicable) - Expanded View Figure Legends (if applicable)

- Corresponding author line paragraph should be removed from p17 - we need the lead author info (such as email) on the title page
- Supplemental material list should be removed from the main ms
- 'S1. FASTA files for KMT5C orthologs.' and 'S2. FASTA files for recombinant proteins' are listed as Files in the ms file, but the actual files are missing
- Please note that the box plots need to be defined in terms of minima, maxima, centre, bounds of box and whiskers, and percentile in the legends of figures 3b; 5e-h; 6c, supplementary figures 3b; 4; 6b.
- Please note that information related to n is missing in the legends of figures 3b; 5e-h; 6c, supplementary figures 3a-b; 4; 6a-b.
- Please note that the error bars are not defined in the legends of supplementary figures 3a; 6a.
- Fig 5E-H seems to show always the same cells in the different panels, is this correct? Please explain this in the figure legend.
- Please consider re-writing the title and abstract based on the final ms. The new findings should be described in present tense in the abstract.

EMBO press papers are accompanied online by A) a short (1-2 sentences) summary of the findings and their significance, B) 2-3 bullet points highlighting key results and C) a synopsis image that is exactly 550 pixels wide and 200-600 pixels high (the height is variable). The synopsis image should provide a sketch of the major findings, like a graphical abstract. Please note that text needs to be readable at the final size. Please send us this information along with the final manuscript.

Referee #2:

In this revised paper, the authors relate how the sequence features drive emergent dynamics and heterochromatin-methylation sensitive behavior of a lysine methyl transferase protein KMT5C. By combining evolutionary analyses, mutant probes, and FRAP analyses, they highlight how motifs embedded in the protein, along with conformational properties of a linker domain, enable partitioning and localization of the KMT5C. Importantly, in their revised study, through careful comparison of accumulation and relative concentrations of molecules when domain-mutants can/cannot partition into chromocenters, they propose a model by which excess HP1 binding sites provide the dynamic scaffold to recruit KMT5C. Furthermore, their model highlights the importance of cooperativity amongst the various motifs uncovered. A major change in this paper is the removal of analogies and experiments to LLPS, which although originally of some interest, do overall help focus the paper on the key findings related to disordered protein properties to heterochromatin recruitment and dynamics of KMT5C.

I report below some remaining concerns, most of which relates to the presentation and interpretation of various mutant and MBP data (both sets of experiments themselves are significant - but the outcomes are not presented a clear and accessible way).

The plotting of partition coefficients time dependence overlaid with representative snaps in Fig.3,5,.. is quite confusing. It would be important to plot separately to understand the differences.

Overall, from the series of expts in Fig 3,5,6 - the authors need to clearly state and clarify the following points:

Mutation of any motif/domain reduces recruitment

however, only mutation of domains in CR1 affect dynamics i.e., one sees much faster recovery (also consistent with stronger loss of localization)

From the MBP recovery experiments, it seems that partition data is not consistently reported in Fig 5 panels and are missing for few mutants upon rescue

In their abstract, the authors write the the first module promotes recruitment via avidity and the second via "optimal" tethering.. perhaps what they mean is that phi3 by itself is a weak recruiter and requires an optimally short and non-negative linker for its effect to be unlocked completely? If this is the case, the authors should clarify appropriately (as its written it seems to imply the second domain does not affect affinity at all). I also recommend they highlight the conformational buffering concept in their

abstract - this is a nice example of this.

Referee #3:

The manuscript by Knechtel et al. dissects the enrichment, dynamics and retention mechanism of the lysine methyltransferase KMT5C (SUV420H2) (and various protein constructs derived from it) in mouse chromocenters that form constitutive heterochromatin compartments. The study combines imaging-based protein enrichment measurements with full/partial/inverse FRAP experiments to characterize KMT5C translocation in/out of chromocenters as well as its mobility within chromocenters. The authors find that the exchange of KMT5C in chromocenters with the surrounding nucleoplasm is slow and on the minute scale. At the same time, the protein exhibits surprisingly rapid mobility and preferred internal mixing within seconds in this compartment. This mobility is indicative of liquid-liquid phase separation (LLPS). However, their subsequent experiments reveal a sustained efficiency of KMT5/HRD chromocenter enrichment over a wide concentration range and saturability of chromocenter binding sites. These latter findings argue against LLPS as a mechanism for KMT5C assembly at chromocenters.

The interaction dynamics of KMT5C are examined in relation to its "heterochromatin retention domain" (HRD) by comparing numerous protein constructs. The authors show that the HRD mediates chromocenter binding via two modules (CR1 and CR2) connected by an intrinsically disordered linker. They demonstrate that the HRD interacts with HP1 proteins via atypical hydrophobic motifs and is critical for chromocenter retention. Using chimeric linker constructs, they show the linker plays a vital role in modulating cooperativity between CR1 and CR2 and acts as a sensor of the chromatin environment. Based on these findings, a mechanistic model is proposed, wherein CR1 uses avidity to enhance binding to HP1/H3K9me3, while CR2 increases local HP1 concentration through tethering. The linker tunes this cooperation in a context-dependent manner.

Overall, this study is well conducted, and the comprehensive mutational analysis provides valuable new insights into the dynamic but tight chromatin interactions of KMT5C and heterochromatin. The results are of broader interest as they elucidate how proteins are retained in chromatin compartments while allowing their dynamic exchange of binding sites within the compartment. I have some points that the authors should consider.

1. As described in the manuscript, several studies have shown that HP1 is highly dynamic. However, FRAP experiments detect an up to 10% immobile fraction of HP1 isoforms. Since KMT5C is much less abundant than HP1, the total concentration of the immobile HP1 fraction in chromocenters could easily match the total concentration of KMT5C. Thus, the statement about the "dramatically different dynamics" between the tight interaction of KMT5C with chromocenters via a highly mobile HP1 might be somewhat misleading.

2. KMT5B (SUV420H1) has significant sequence and structural similarity to KMT5C and is also targeted by HP1 to chromocenters. In FRAP measurements, KMT5B shows a significantly faster recovery upon chromocenter photobleaching. Can this be explained by differences in the HRD domain and the model put forward in Fig. 7?

3. KMT5C might be able to interact with multiple HP1 proteins since within the HRD, the CR1 and CR2 modules could independently interact with HP1. This could be relevant for a hopping mechanism involving transient binding to two different parts of the nucleosome chain.

4. The quantitative FRAP analysis (Methods, p11 top) has some issues. The equation for the monoexponential $f(t) = A(1 - e^{-t/\tau})$ fit does not make sense and presumably should read $f(t) = A(1 - e^{-t/\tau D})$ with τD being the characteristic diffusion time. Using the equations $D_{app} = r^2/t$ with t being the "full fluorescence recovery" seems questionable since this time point is not well defined in the experiments, e.g., red curves for total FRAP in Fig. S3 and S6 (in ref. 77, t at 80% recovery is used, which seems arbitrary). Instead, $D_{app} = r^2/(4 \cdot t_{1/2})$ with $t_{1/2}$ being the recovery half-time, would be a well-established approximation (Axelrod, Biophys J, 1976). Furthermore, it appears problematic to use a diffusion model for total FRAP of KMT5C since its slow recovery would be better described by koff from a binding or reaction-diffusion model (Sprague, Biophys J, 2004). Thus, the authors should make clear that their analysis yields empirical parameters for comparing the mobility of different constructs in FRAP experiments rather than meaningful (apparent) diffusion coefficients. For the comparison, a supplementary table with FRAP-derived parameters would be helpful.

5. In the rebuttal letter, the authors state that the new data that are included in the revised manuscript would be consistent with a binding and transfer/hopping mechanism proposed by Reviewer #1. In the discussion and Fig. 7, the focus is on the molecular mechanism that drives chromatin interactions of KMT5C via CR1 and CR2. However, it remains unclear which mechanism the authors propose to rationalize the rapid translocation of KMT5C within chromocenters since a hopping/transfer mechanism is not mentioned in the discussion of the manuscript.

Minor points

To avoid confusion, I would introduce the different gene names (KMT5B/C and SUV420H1/2, CBX1/3/5 and HP1 $\beta/\gamma/\alpha$ and KMT1A/B and SUV39H1/2) in the introduction and then consistently use one name.

Response to reviews for EMBOR-2022-54786V2

The summary provided by each referee has been deleted and responses to each point are provided below. Referee comments are in bold and our responses immediately follow. Corresponding text changes are highlighted in yellow in the manuscript document.

Referee 2

The plotting of partition coefficients time dependence overlaid with representative snaps in Fig.3, 5, is quite confusing. It would be important to plot separately to understand the differences. The figures have been modified as requested. We removed the overlays and now plot this summary data separately. In addition, full data in each case is provided in the supplementary section.

Overall, from the series of experiments in Fig 3, 5, and 6, the authors need to clearly state and clarify the following points:

Mutation of any motif/domain reduces recruitment. However, only mutation of domains in CR1 affect dynamics (i.e., one sees much faster recovery and also consistent with stronger loss of localization). This is largely true but we do see effects from mutations in CR2. We have clarified the text to emphasize this and improve clarity. This is also evident in figure 3c where multiple dynamic and partitioning parameters are hierarchically clustered, showing the CR2 mutations form a separate group. With the changes suggested for figure layout, this is more evident now. At multiple points in the manuscript, we have emphasized that CR2 makes important contributions to localization and dynamics. This is also noted in the abstract (first highlighted sentence).

From the MBD recovery experiments, it seems that partition data is not consistently reported in Fig 5 panels and are missing for few mutants upon rescue. As noted above for figure 3, the overlaid data has been re-organized for clarity. In addition, all related partition coefficient and FRAP data is provided as a supplemental figure.

In their abstract, the authors write the first module promotes recruitment via avidity and the second via "optimal" tethering. Perhaps what they mean is that phi3 by itself is a weak recruiter and requires an optimally short and non-negative linker for its effect to be unlocked completely? If this is the case, the authors should clarify appropriately (as its written it seems to imply the second domain does not affect affinity at all). I also recommend they highlight the conformational buffering concept in their abstract - this is a nice example of this. Text has been modified to incorporate this suggestion and improve clarity on motif use. In addition, an explicit mention of conformational buffering is provided in the abstract.

Referee 3

Major points

1. As described in the manuscript, several studies have shown that HP1 is highly dynamic. However, FRAP experiments detect an up to 10% immobile fraction of HP1 isoforms. Since KMT5C is much less abundant than HP1, the total concentration of the immobile HP1 fraction in chromocenters could easily match the total concentration of KMT5C. Thus, the statement about the "dramatically different dynamics" between the tight interaction of KMT5C with chromocenters via a highly mobile HP1 might be somewhat misleading. We agree that at endogenous proteins levels, the small immobile HP1 population could match the total concentration of KMT5C. We emphasize this stoichiometry difference in the discussion, where we suggest that the vast excess of HP1 can support retention of KMT5C even when it is over-expressed, such that it may act as a 'preferred HP1 client.' These points formulate our concluding statement in the abstract: 'we show that KMT5C has evolved a robust tethering strategy that uses minimal sequence determinants to harness highly dynamic HP1 proteins for retention within heterochromatin compartments.' The findings also suggest KMT5C could be an important determinant of the HP1 immobile fraction. For the sentence the reviewer has mentioned, we have modified it to read 'dramatically different bulk dynamics,' which occurs in the second paragraph on page 2.

2. KMT5B (SUV420H1) has significant sequence and structural similarity to KMT5C and is also targeted by HP1 to chromocenters. In FRAP measurements, KMT5B shows a significantly faster recovery upon chromocenter photobleaching. Can this be explained by differences in the HRD domain and the model put forward in Fig. 7? There is no obvious homology between the regions of KMT5B and KMT5C that control localization and have added a sentence to indicate this fact. We have mapped determinants for KMT5B localization as part of a separate study and, while also dependent on HP1, it uses a very different strategy. Our preference is to omit this data because it would impact our ability to publish this work. A statement regarding differences between KMT5B and KMT5C appears in the third paragraph on page 3.

3. KMT5C might be able to interact with multiple HP1 proteins since within the HRD, the CR1 and CR2 modules

could independently interact with HP1. This could be relevant for a hopping mechanism involving transient binding to two different parts of the nucleosome chain. This was brought up in the original review and we have modified to the last paragraph of the discussion to include this a possible mechanism. This now includes an explicit mention of hopping or transfer, as well as facilitated dissociation. Relevant statements are highlighted in the last paragraph on page 10 (which continues on page 11).

4. The quantitative FRAP analysis (Methods, p11 top) has some issues. The equation for the monoexponential $f(t) = A(1 - e^{-t/\tau D})$ fit does not make sense and presumably should read $f(t) = A(1 - e^{-t/\tau D})$ with τD being the characteristic diffusion time. Using the equations $D_{app} = r^2/t$ with t being the "full fluorescence recovery" seems questionable since this time point is not well defined in the experiments, e.g., red curves for total FRAP in Fig. S3 and S6 (in ref. 77, t at 80% recovery is used, which seems arbitrary). Instead, $D_{app} = r^2/(4 \cdot t_{1/2})$ with $t_{1/2}$ being the recovery half-time, would be a well-established approximation (Axelrod, Biophys J, 1976). Furthermore, it appears problematic to use a diffusion model for total FRAP of KMT5C since its slow recovery would be better described by koff from a binding or reaction-diffusion model (Sprague, Biophys J, 2004). Thus, the authors should make clear that their analysis yields empirical parameters for comparing the mobility of different constructs in FRAP experiments rather than meaningful (apparent) diffusion coefficients. For the comparison, a supplementary table with FRAP-derived parameters would be helpful. Based on the suggestion of this reviewer that we should focus on parameters derived from comparative FRAP analyses rather than the apparent diffusion coefficient of a single protein, we have made the following changes. First, we have removed the spot bleach data from Figure 1 because we agree that, in isolation, it does not enable meaningful comparisons. Second, we now include mobile fractions for the full series of mutant proteins in figure 3c. Last, we agree that slow recovery would be better described by reduced K_{off} , which is also consistent with the modulation of C_{eff} by intrinsically disordered linkers. We have added a sentence to this effect in the discussion, which is highlighted in the first paragraph on page 10.

5. In the rebuttal letter, the authors state that the new data that are included in the revised manuscript would be consistent with a binding and transfer/hopping mechanism proposed by Reviewer #1. In the discussion and Fig. 7, the focus is on the molecular mechanism that drives chromatin interactions of KMT5C via CR1 and CR2. However, it remains unclear which mechanism the authors propose to rationalize the rapid translocation of KMT5C within chromocenters since a hopping/transfer mechanism is not mentioned in the discussion of the manuscript. As noted in the response to point 3, we have now added this content to the discussion. This was brought up as an alternative to phase separation to account for the diffusive behavior of KMT5C through either hopping or transfer via facilitated dissociation. We also discuss this together with how the bipartite structure may support this behavior through alternative high and low affinity states. Addressed in point 3.

Minor points

To avoid confusion, I would introduce the different gene names (KMT5B/C and SUV420H1/2, CBX1/3/5 and HP1 β/γ and KMT1A/B and SUV39H1/2) in the introduction and then consistently use one name. Agree and text modified accordingly. We have included the information requested by the reviewer at first mention of each protein in the manuscript. For consistency, we use the official gene and protein nomenclature for each protein, but refer generically to HP1 when referring to the general family of proteins. This is highlighted at the top of page 2 and in the third paragraph on page 3 where KMT5B is first mentioned.

Dr. D Underhill
University of Alberta
Oncology, Medical Genetics
11560 University Avenue NW
Edmonton, Alberta T6G 1Z2
Canada

Dear Alain,

I am very pleased to accept your manuscript for publication in the next available issue of EMBO reports. Thank you for your contribution to our journal.

Referee #3:

The authors have addressed my comments in the revised manuscript. The changes are well reflected in both the text and figures, with thoughtful additions to the discussion and clarification of technical points. The manuscript provides valuable mechanistic insights into how minimal sequence features can achieve robust protein compartmentalization through conformational buffering.
